# Diblock dialternating terpolymers by one-step/one-pot highly selective organocatalytic multimonomer polymerization

Jiaxi Xu[1], Xin Wang 🔟 [1] & Nikos Hadjichristidis 🔟 [1✉]

The synthesis of well-defined block copolymers from a mixture of monomers without additional actions ("one-pot/one-step") is an ideal and industrially valuable method. In addition, the presence of controlled alternating sequences in one or both blocks increases the structural diversity of polymeric materials, but, at the same time, the synthetic difficulty. Here we show that the "one-pot/one-step" ring-opening terpolymerization of a mixture of three monomers (N-sulfonyl aziridines; cyclic anhydrides and epoxides), with *tert*-butylimino-tris(dimethylamino)phosphorene (*t*-BuP$_1$) as a catalyst, results in perfect diblock dialternating terpolymers having a sharp junction between the two blocks, with highly-controllable molecular weights and narrow molecular weight distributions (Đ < 1.08). The organocatalyst switches between two distinct polymerization cycles without any external stimulus, showing high monomer selectivity and kinetic control. The proposed mechanism is based on NMR, in-situ FTIR, SEC, MALDI-ToF, reactivity ratios, and kinetics studies.

---

[1] Polymer Synthesis Laboratory, Physical Sciences and Engineering Division, KAUST Catalysis Center, King Abdullah University of Science and Technology (KAUST), Thuwal 23955, Saudi Arabia. ✉email: nikolaos.hadjichristidis@kaust.edu.sa

Block copolymerization technology has attracted considerable interest in commercial applications as it combines in the same molecule the unique characteristics of each segment[1–4]. By controlling the composition, molecular weight/ molecular weight distribution, and structure of each segment, the properties of the final block copolymers can be adjusted[5]. This is why block copolymers are widely used in thermoplastic elastomer[6,7], drug delivery[8], membrane[9], lithography[10], and mesoporous material[11] industrial sectors. The synthesis of well-defined block copolymers is achieved using a living or controlled/ living polymerization to ensure predictable molecular weight/ structure and a high degree of molecular, structural, and compositional homogeneity.

Diblock copolymers via a living/controlled polymerization can be synthesized by (a) "two-pot/two-step", (b) "one-pot/two-step", and (c) "one-pot/one-step" procedures. In the "two-pot/two-step" procedure, the first segment synthesized in one reactor requires complex and costly modification/purification processes before being used for the second polymerization in another reactor. The "one-pot/two-step" procedure avoids the isolation of the first block and requires either a perfect living polymerization (stability of the living chains and complete conversion of monomer to polymer) compatible with both monomers, as is the case of anionic polymerization, or a switching procedure after the total consumption of the first monomer, such as catalyst switch (Lewis acid/base-[12,13] or redox-[14,15] switchable), or experimental condition switch (thermal[16,17], electricity[18], photochemistry[19,20], and gas atmosphere[21,22]). Switching between two polymerizations is both critical and challenging; an early switch leads to a gradient/ random sequence, and a delayed switch can cause other harmful side reactions (such as transesterification[23] and decomposition[24]). If an auto-switch occurs after the complete consumption of the first monomer, we have the "one-pot/one-step" procedure, and thus the synthesis of diblock copolymers is simple and industrially valuable. A few diblock copolymers are synthesized by the "one-pot/one-step" procedure due to strict requirements regarding monomer reactivity ratios and activity/selectivity of initiating/ catalytic system. Another way to "one-pot/one-step" synthesis of block copolymers requires heterofunctional initiators for polymerizations that can proceed simultaneously[25–27].

On the other hand, the "one-pot/one-step" alternating copolymerization of two monomers is an excellent, but challenging method to synthesize polymers with a rich diversity in structures and properties[28–30]. The following monomer pairs of epoxide/ $CO_2$[31–33], epoxide/cyclic anhydride[28,34], epoxide/isocyanate[35,36], and aziridine/cyclic anhydride[37,38], resulted in polycarbonates, polyesters, polyurethanes, and poly(ester amide)s, respectively. However, the synthesis of diblock copolymers with two controlled alternating sequences in both blocks with a sharp boundary by the "one-pot/one-step" procedure is extremely challenging.

In 2008, Coates and co-workers were the first to report the terpolymerization of cyclohexene oxide (CHO), diglycolic anhydride, and carbon dioxide towards diblock dialternating terpolymers using metal catalysts[39], but, unfortunately like a few others[40–43], the obtained dialternating polyester-b-polycarbonate terpolymers, were contaminated by about 10% of a tapered structure. It is well known that the tapered chain between the two neat blocks affects their mechanical properties[44].

In this work, we extend our previous work on the ring-opening copolymerization (ROCOP) of N-sulfonyl aziridines and cyclic anhydrides leading to either alternating copolymers or (alternating copolymer)-b-homopolymer[37], to ring-opening terpolymerization of N-sulfonyl aziridines, cyclic anhydrides, and epoxides. We discovered that tert-butylimino-tris(dimethylamino)phosphorene (t-BuP$_1$) catalyst shows high selectivity in monomers and kinetic control for two alternating

copolymerization cycles without competitive side reactions leading to perfect diblock dialternating poly(ester amide)-b-polyester via a "one-pot/one-step" with a sharp boundary, controlled molecular weight and narrow molecular weight distribution.

## Results

**Terpolymerization of N-tosylaziridine (TAz), phthalic anhydride (PA), and epoxide.** As a starting point for the current investigations, the terpolymerization of TAz, PA, and propylene oxide (PO) with the initiator [BnN(H)Ts] without catalyst was performed at 100 °C in tetrahydrofuran (THF) or bulk (entries 1 and 2 in Table 1 and Fig. 1). No polymer signals were detected by proton nuclear magnetic resonance ($^1$H NMR) spectroscopy, indicating the absence of any polymerization. t-BuP$_1$ was then selected as the catalyst for the terpolymerization because, even in small amounts, it catalyzes the controlled ROCOP of TAz/PA[37] and PO/PA[45]. After the addition of t-BuP$_1$ ([BnN(H)Ts]/[t-BuP$_1$] = 1/0.5), the ROCOP of TAz/PA started immediately (entry 3 in Table 1, Supplementary Table 1, and Supplementary Fig. 1) as evidenced by the $^1$H NMR spectra of samples (Supplementary Fig. 1) collected during the terpolymerization. The aromatic peaks (7.81 and 7.33 p.p.m.) decreased rapidly with the simultaneous increase of the methylene signals of poly(TAz-alt-PA) (4.42 and 4.19 p.p.m.). TAz and PA conversions reached 91 and 30% within 45 min ([TAz]$_0$/[PA]$_0$ = 1/3), respectively. The characteristic signals of poly(PA-alt-PO) (5.41 p.p.m.) and poly(propylene oxide) (PPO, 3.30–3.60 p.p.m.) were not observed in the $^1$H NMR spectra (Supplementary Fig. 1), indicating that insertion of PO was completely suppressed. The size-exclusion chromatography (SEC, Supplementary Fig. 1) traces were shifted to high molecular weight, but maintained unimodal/narrow distributions ($Đ < 1.07$), supporting that ROCOP of TAz/PA is well controlled without any transesterification reaction. Since ROCOP of TAz/PA has a first-order dependence on the TAz concentration[37], slow copolymerization rates were observed at high TAz conversions (Supplementary Fig. 1 and Supplementary Table 1). Complete TAz consumption was reached in 1.5 h, and interestingly, without any PO insertion into the chain. The signal from the second-stage alternating units in poly(PA-alt-PO) (5.41 p.p.m.) was observed by $^1$H NMR only after 5 h (Supplementary Fig. 1). There is no overlap between the two copolymerization cycles, indicating the absence of a tapered sequence at the junction between the two alternating blocks. However, the SEC traces gradually widen with increasing PA conversion in the second copolymerization cycle and eventually split into two peaks, the lower corresponding to the unreacted first block (Supplementary Fig. 1).

Then, experiments under the same feed ratios were carried out in bulk (entries 5 and 6 in Table 1, Supplementary Table 2, and Supplementary Fig. 2). The increased monomer concentration accelerates the ROCOP of TAz/PA, with the total TAz consumption reached at 30 min (Supplementary Fig. 2). SEC traces and $^1$H NMR spectra show that the first-segment poly(TAz-alt-PA) is well defined with narrow molecular weight distributions ($Đ < 1.06$) without any PO insertion. However, during the second copolymerization cycle of PA/PO, the same problem was observed: the SEC traces are split into two peaks (Supplementary Fig. 2), with molecular weight distributions of each peak remaining narrow (1.03 and 1.05). The increased monomer concentration only accelerates the two copolymerization rates, but the bimodal peaks on SEC traces are consistent with the previous ones. The low molecular weight peaks appear at the same retention volume as the one of the first-segment poly(TAz-alt-PA) (macroinitiator) with almost the same molecular weight distributions (Supplementary Figs. 1 and 2),

**Table 1 Terpolymerization of TAz, PA, and epoxides using $t$-BuP$_1$ as a catalyst[a].**

| Entry | Epoxides | Time (h) | TAz conv.[b] (%) | PA conv.[b] (%) | $M_{n,theo}$[c] (kg mol$^{-1}$) | $M_{n,NMR}$[b] (kg mol$^{-1}$) | Đ[d] |
|---|---|---|---|---|---|---|---|
| 1[e] | PO | 24 | — | — | — | — | — |
| 2[f] | PO | 24 | — | — | — | — | — |
| 3[g] | PO | 1.5 | 99 | 33 | 10.5 | 11.2 | 1.05 |
| 4[g] | | 21 | 99 | 94 | 21.9 | 23.6 | 1.04; 1.05[h] |
| 5[i] | PO | 0.5 | 99 | 33 | 10.5 | 11.2 | 1.05 |
| 6[i] | | 4 | 99 | 99 | 22.8 | 24.2 | 1.03; 1.05[h] |
| 7 | PO | 0.16 | 99 | 33 | 10.5 | 11.2 | 1.03 |
| 8 | | 5 | 99 | 99 | 22.8 | 24.8 | 1.03 |
| 9[j] | PO | 1 | 99 | 33 | 10.5 | 11.2 | 1.05 |
| 10[j] | | 6 | 99 | 87 | 20.6 | 21.7 | 1.06 |
| 11 | EO | 0.25 | 99 | 33 | 10.5 | 11.2 | 1.03 |
| 12 | | 2 | 99 | 80 | 18.7 | 19.7 | 1.05 |
| 13 | | 3 | 99 | 99 | 22.0 | 24.0 | 1.08 |
| 14[k] | EO | 1.5 | 99 | 33 | 10.5 | 11.2 | 1.04 |
| 15[k] | | 30 | 99 | 99 | 22.0 | 24.2 | 1.03 |
| 16 | BO | 0.5 | 99 | 33 | 10.5 | 11.2 | 1.03 |
| 17 | | 5 | 99 | 99 | 23.6 | 25.1 | 1.04 |
| 18[l] | PO | 0.25 | 99 | 33 | 10.4 | 11.2 | 1.04 |
| 19[l] | | 5 | 99 | 94 | 21.8 | 23.8 | 1.10 |
| 20[m] | PO | 0.25 | 99 | 33 | 18.3 | 19.6 | 1.02 |
| 21[m] | | 3 | 99 | 99 | 30.5 | 32.0 | 1.03 |
| 22[n] | PO | 0.25 | 99 | 33 | 10.5 | 11.2 | 1.03 |
| 23[n] | | 3 | 99 | 99 | 22.8 | 25.6 | 1.77 |
| 24[o] | PO | 0.25 | 99 | 33 | 10.5 | 11.2 | 1.06 |
| 25[o] | | 4 | 99 | 99 | 22.8 | 24.8 | 1.25 |
| 26[p] | PO | 2 | 99 | 99 | 22.8 | 6.12 | 1.29 |
| 27[q] | PO | 7 | 99 | 99 | 22.8 | 7.78 | 1.38 |
| 28[r] | PO | 0.25 | 99 | 49 | 10.5 | 11.2 | 1.03 |
| 29[r] | | 4 | 99 | 99 | 16.7 | 18.2 | 1.04 |
| 30[s] | PO | 0.25 | 99 | 24 | 10.5 | 11.2 | 1.03 |
| 31[s] | | 8 | 99 | 99 | 28.9 | 30.0 | 1.02 |

[a]The terpolymerizations were performed at a ratio of [TAz]$_0$/[PA]$_0$/[BnN(H)Ts]$_0$/[$t$-BuP$_1$]$_0$ = 30/90/1/1 at 100 °C ([TAz]$_0$ = 1.0 M in epoxides).
[b]Determined by $^1$H NMR.
[c]Calculated as follows: (MW of I) + ([TAz]$_0$/[I]$_0$) × conv.(TAz) × (MW of TAz + MW of PA) + {([PA]$_0$/[I]$_0$) × conv.(PA) − ([TAz]$_0$/[I]$_0$) × conv.(TAz)} × (MW of epoxides + MW of PA).
[d]Determined by SEC at 35 °C in THF (1.0 mL min$^{-1}$) using PSt standards.
[e]Without catalysts ([TAz]$_0$ = 0.5 M in epoxides and THF).
[f]Without catalysts in bulk ([TAz]$_0$ = 1.0 M in epoxides).
[g][TAz]$_0$/[PA]$_0$/[BnN(H)Ts]$_0$/[$t$-BuP$_1$]$_0$ = 30/90/1/0.5 in epoxides and THF ([TAz]$_0$ = 0.5 M in epoxides and THF, the volume amount of epoxides is same as THF).
[h]Bimodal peaks.
[i][TAz]$_0$/[PA]$_0$/[BnN(H)Ts]$_0$/[$t$-BuP$_1$]$_0$ = 30/90/1/0.5 ([TAz]$_0$ = 1.0 M in epoxides).
[j][TAz]$_0$/[PA]$_0$/[BnN(H)Ts]$_0$/[$t$-BuP$_1$]$_0$ = 30/90/1/1 ([TAz]$_0$ = 2.0 M in epoxides).
[k]At 60 °C.
[l]Initiated by 1,4-benzenedimethanol.
[m]Initiated by polyethylene glycol.
[n]Catalyzed by $t$-BuP$_4$.
[o]Catalyzed by $t$-BuP$_2$.
[p]Catalyzed by DBU.
[q]Catalyzed by TBD.
[r][TAz]$_0$/[PA]$_0$/[BnN(H)Ts]$_0$/[$t$-BuP$_1$]$_0$ = 30/60/1/1.
[s][TAz]0/[PA]0/[BnN(H)Ts]0/[$t$-BuP1]0 = 30/120/1/1.

**Fig. 1 Terpolymerizations of TAz, PA, and PO.** The synthesis of diblock dialternating terpolymers with a sharp boundary instead of a tapered sequence at the conjunction between the two alternating blocks.

suggesting that part of it cannot participate in the second copolymerization cycle. The diffusion-ordered spectroscopy (DOSY) NMR spectrum of the precipitated polymer (Supplementary Fig. 3) shows two diffusion coefficients, corresponding to poly(TAz-*alt*-PA)-*b*-poly(PA-*alt*-PO) and unreacted poly(TAz-*alt*-PA) and thus supporting the SEC results.

The unreacted first-segment poly(TAz-*alt*-PA) may be the cause of insufficient $t$-BuP$_1$. Thus, an equivalent $t$-BuP$_1$ to BnN(H)Ts was then used for the terpolymerization of TAz, PA, and PO in bulk (entries 7 and 8 in Table 1, Fig. 2, and Supplementary Table 3). The increased $t$-BuP$_1$ concentration accelerates the rates of both copolymerization cycles. The

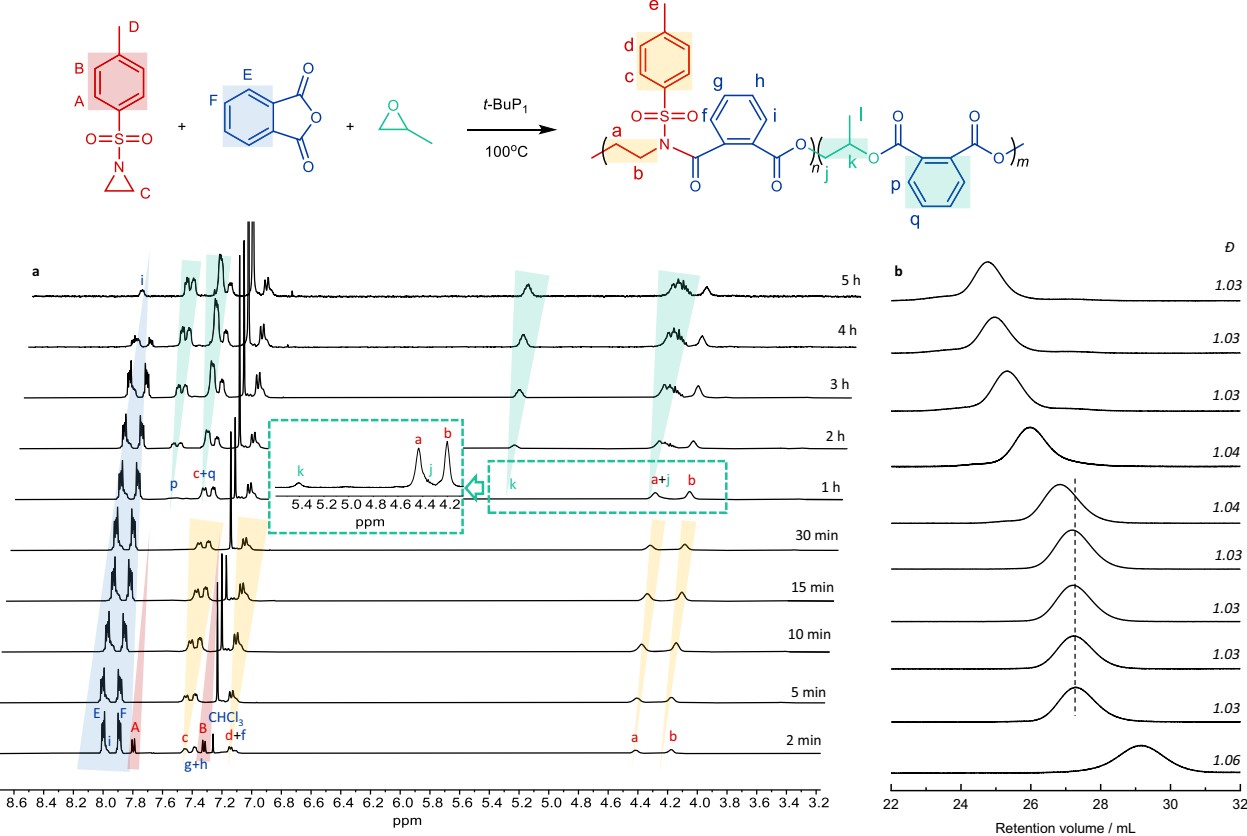

**Fig. 2 Terpolymerization of TAz, PA, and PO at different reaction times. a** Characteristic signals monitored by [1]H NMR spectra. **b** Retention volume monitored by SEC traces (entries 7 and 8 in Table 1 and Supplementary Table 3).

ROCOP of TAz/PA was completed quickly within 10 min, and after an additional 50 min, the second ROCOP of PA/PO was started (Fig. 2a and Supplementary Table 3). Therefore, there is no overlap between the two copolymerization cycles, leading to a sharp boundary instead of a tapered sequence at the conjunction between the two alternating blocks. In addition, no ether (3.30–3.60 p.p.m.) or amine linkage (3.20–3.30 p.p.m.) was observed, indicating the absence of TAz and PO self-homopolymerization. Narrow and symmetric SEC traces were observed during the two alternating copolymerization cycles (Fig. 2b, Đ < 1.06), indicating that the entire first living alternating block participated in the second copolymerization. All results reveal that increased $t$-BuP$_1$ concentration accelerates both copolymerization rates and activates the entire first block (macroinitiator) for the second copolymerization without inserting any PO into the first-stage copolymerization.

**Chain microstructure of poly(TAz-*alt*-PA)-*b*-poly(PA-*alt*-PO).** FTIR was used to confirm the structure of the first- and second-stage copolymers. After complete consumption of TAz, the first-stage copolymer was precipitated in methanol for FTIR characterization (Fig. 3a). The absorption peaks at 1721 and 1691 cm$^{-1}$ correspond to the stretching vibration of C=O in esters and of amide groups, respectively, whereas at 1276 and 1252 cm$^{-1}$ to the stretching vibration of C–N and C–O, respectively. The results confirm the presence of ester and amide linkages in the first-stage copolymer, indicating the presence of an alternating sequence of TAz and PA. After complete consumption of PA, the final diblock dialternating terpolymer was isolated and characterized by FTIR spectroscopy (Fig. 3b). The stretching

vibration of C=O at 1721 and C–O at 1252 cm$^{-1}$ in ester groups become significantly stronger (Fig. 3a, b, e), confirming the alternating sequence of PA and PO in the second stage.

The chemical structures of the diblock dialternating terpolymers were further analyzed by [1]H NMR and [13]C{[1]H} NMR spectroscopy (Fig. 4 and Supplementary Fig. 5). The characteristic signal peaks of the methylene protons of poly(TAz-*alt*-PA) between ester and amide groups were observed at 4.44 and 4.21 p.p.m. (a and b in Fig. 4a). We did not observe any amine linkage signal at 3.15–3.34 p.p.m., indicating the perfect alternating copolymerization of TAz/PA. The characteristic signal peaks of the second-stage copolymer poly(PA-*alt*-PO) were observed at 5.41 and 4.37 p.p.m., corresponding to the methine and methylene signals, respectively (j and k in Fig. 4a). There are no ether linkage signals at 3.30–3.60 p.p.m., indicating that there is no PO self-propagation. The carbonyl regions of the [13]C{[1]H} NMR spectrum (Supplementary Fig. 5) at 169.9 and 164.7 p.p.m. represent the amide and ester groups of poly(TAz-*alt*-PA), respectively. The characteristic signal peaks of the carbonyl group from poly(PA-*alt*-PO) were observed at 167.0 and 168.8 p.p.m. The absence of any other peaks of heterosequences excludes the presence of tapered sequence and transesterification reactions during terpolymerization.

The final diblock dialternating terpolymer was also analyzed by DOSY NMR spectra (Fig. 4b), showing a single diffusion coefficient corresponding to the product and not either to a blend of poly(TAz-*alt*-PA) and poly(TAz-*alt*-PA)-*b*-poly(PA-*alt*-PO) (Supplementary Fig. 3) or poly(TAz-*alt*-PA) and poly(PA-*alt*-PO) (Supplementary Fig. 4). The DOSY NMR spectroscopy supports the sharp connection of the two copolymer segments.

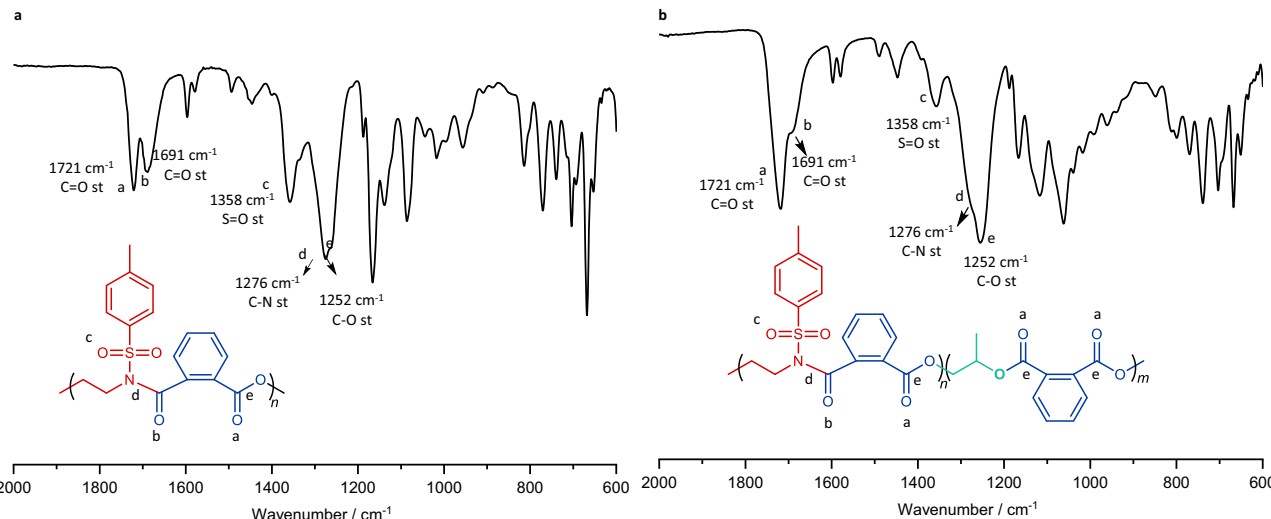

**Fig. 3 Fourier-transform infrared spectroscopy (FTIR) characterization. a** FTIR spectra of poly(TAz-*alt*-PA). **b** FTIR spectra of poly(TAz-*alt*-PA)-*b*-poly(PA-*alt*-PO).

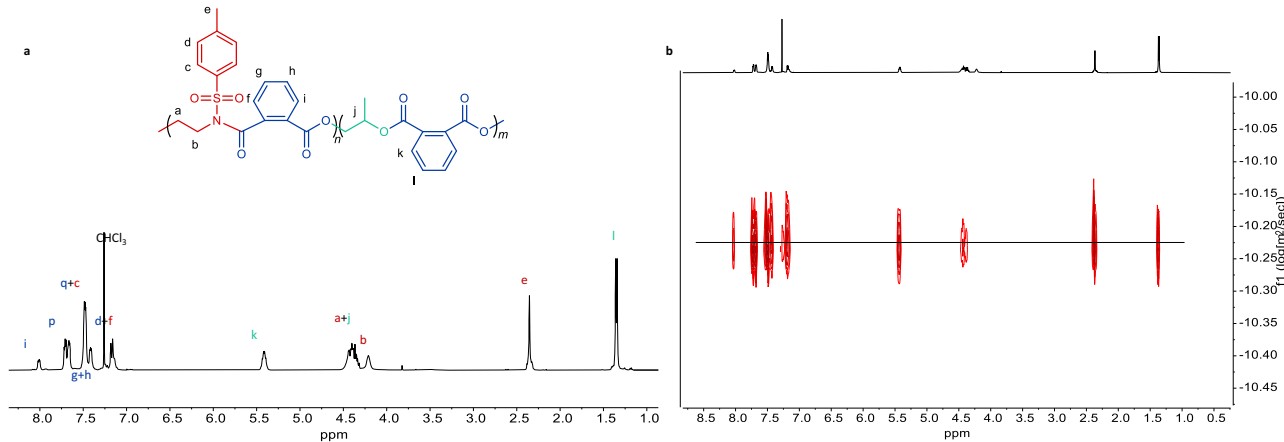

**Fig. 4 Nuclear magnetic resonance characterization. a** $^1$H NMR of poly(TAz-*alt*-PA)-*b*-poly(PA-*alt*-PO). **b** DOSY NMR spectra of poly(TAz-*alt*-PA)-*b*-poly(PA-*alt*-PO).

Matrix-assisted laser desorption ionization time-of-flight mass spectroscopy (MALDI-ToF MS) was used to confirm further the structure and terminal functional group of the first-stage poly(TAz-*alt*-PA) (Fig. 5). The first-stage copolymer was obtained after complete TAz conversion at a low molar ratio TAz:PA = 15:45 in PO. The two series of peaks obtained correspond to the sodium- or potassium-cationized poly(TAz-*alt*-PA), containing BnN(H)Ts at one chain end and TAz or PA at the other chain end. The molecular mass of the adjacent peaks shows a fixed interval of 345.07, which is the molecular mass of a repeating unit of TAz and PA. The results show that the copolymers have a perfectly alternating unit of TAz and PA, indicating the absence of any transesterification side reaction during terpolymerization. There is no insertion of PO into the copolymer, indicating that there is no tapered sequence between the two copolymerization cycles. All analytical and spectroscopy techniques support the formation of the desired diblock dialternating terpolymer poly(TAz-*alt*-PA)-*b*-poly(PA-*alt*-PO) without any tapered sequences.

**Kinetic analysis**. The kinetics of the terpolymerization of TAz, PA, and PO was also studied by NMR and in-situ FTIR. Since the low-boiling point of PO (34 °C) easily causes changes in the reaction concentration, we conducted a series of parallel experiments monitored by $^1$H NMR spectra by quenching aliquots for convincing experimental results (repeat all results three times). As shown in Fig. 6a, the first-stage copolymerization of TAz and PA (Supplementary Table 3) is very fast, with 97% TAz and 32% PA conversion in 5 min as monitored by $^1$H NMR ([TAz]$_0$/[PA]$_0$ = 1/3). The molecular weight of the copolymer calculated by $^1$H NMR increases with the reaction time, in agreement with the theoretical molecular weight of poly(TAz-*alt*-PA) (Fig. 6b). All the SEC traces are symmetrical, monomodal, and narrow (Fig. 6b, Đ < 1.06). These results suggest a living copolymerization behavior. After TAz was completely consumed, the second-stage copolymerization of PA and PO proceeded (Fig. 6a). A linear relationship is observed between the reaction time and the PA conversion, indicating a zero-order dependence on the PA concentration, which is consistent with previously reported work[46]. The molecular weight of the copolymers against reaction time shows a linear increase in the second stage, and the diblock dialternating terpolymers maintain the narrow molecular weight distribution (Fig. 6b, Đ < 1.04).

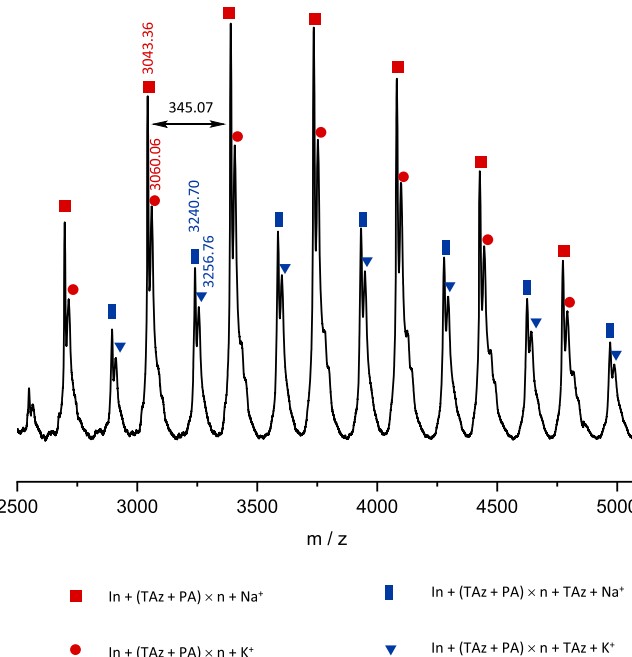

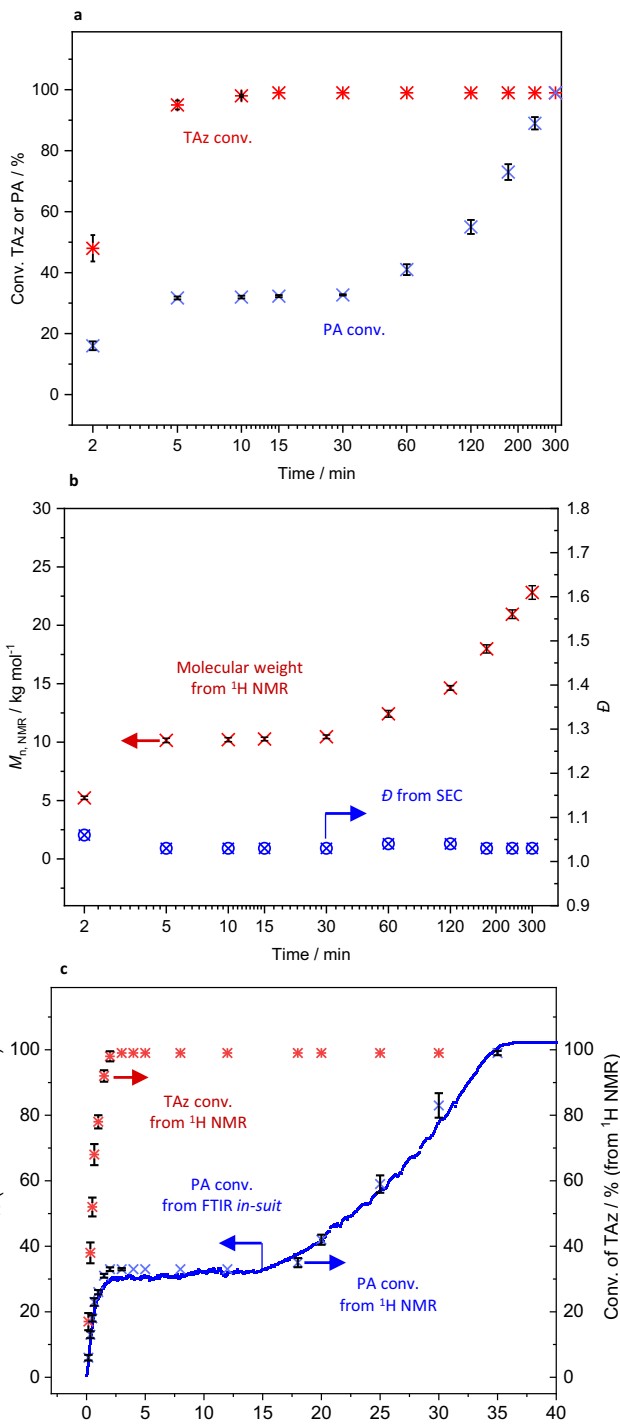

**Fig. 5 MALDI-ToF characterization.** MALDI-ToF MS spectrum of the first-stage copolymer poly(TAz-*alt*-PA).

In order to investigate the terpolymerization behavior, a study was performed at 60 °C under the same conditions and monitored by in situ FTIR and $^1$H NMR (Fig. 6c). The decreased temperature slows down the reaction rates of both the first-stage ($k_{\text{obs,first stage}} = 11.7196 \text{ mol L}^{-1}\text{ h}^{-1}$ in 100 °C; $k_{\text{obs,first-stage}} = 0.4773 \text{ mol L}^{-1}\text{ h}^{-1}$ in 60 °C) and the second-stage copolymerization ($k_{\text{obs,second stage}} = 0.4500 \text{ mol L}^{-1}\text{ h}^{-1}$ in 100 °C; $k_{\text{obs,second stage}} = 0.0682 \text{ mol L}^{-1}\text{ h}^{-1}$ in 60 °C) (Supplementary Figs. 45–48). Monomers conversions by in situ FTIR are consistent with those of $^1$H NMR. After complete consumption of TAz, a dormant period is observed by both in situ FTIR and $^1$H NMR spectroscopy. The trace amount of TAz in the reaction solution may suppress the insertion of PO. Only after all TAz has been converted to copolymers does the second copolymerization of PA/PO begin. In order to verify the relationship between TAz and the dormant period, another part of TAz was fed during the second stage of the copolymerization (PA/PO ROCOP) (Supplementary Figs. 43 and 49 and Supplementary Table 20). It was found that the ROCOP of PA/PO was immediately transferred to ROCOP of TAz/PA. Another new dormant period was observed after the total consumption of TAz, suggesting that residual TAz inhibits the ROCOP of PA/PO. As a result, no tapered sequence is observed at the junction of the two block copolymers.

**Proposed terpolymerization mechanism**. Based on the above experiments, two plausible copolymerization pathways are proposed (Fig. 7 and Supplementary Fig. 50). The two alternating blocks come from the two alternating copolymerization cycles: TAz/PA ROCOP and PA/PO ROCOP, with PA to participate in both copolymerization cycles. In the first cycle, TAz/PA ROCOP takes place, and kinetic studies show that the PA ring-opening is fast. The resulting phosphazenium carboxylate intermediate ($-\text{COO}^-/t\text{-BuP}_1\text{-H}^+$, Supplementary Fig. 51) from PA reacts with TAz instead of PO. After ring opening of TAz, the proton on the $t$-BuP$_1$-H$^+$ is transferred to the new chain end, forming $-\text{N(Ts)H}/t$-BuP$_1$ (Supplementary Fig. 53), due to the stronger basicity of

**Fig. 6 Polymerization kinetic analysis. a** Plots of monomer conversions versus time (100 °C) monitored by $^1$H NMR spectroscopy. **b** $M_{\text{n, NMR}}$ and $Đ$ versus time. **c** Plots of monomer conversions versus time (60 °C) monitored by $^1$H NMR spectroscopy and in situ FTIR (error bars represent the standard error).

$-\text{N(Ts)}^-$. The generated amine $-\text{N(Ts)H}$ rapidly attacks PA instead of PO (Fig. 7, cycle 1). The fast proton exchange between the $t$-BuP$_1$ and $t$-BuP$_1$-H$^+$ is plausible and supported by Supplementary Fig. 53[47]. TAz/PA ROCOP proceeds quickly until TAz is almost completely consumed. At the end-point of the first copolymerization cycle, slow copolymerization rates were observed at high TAz conversions, and finally, a dormant period. The $t$-BuP$_1$/$t$-BuP$_1$-H$^+$

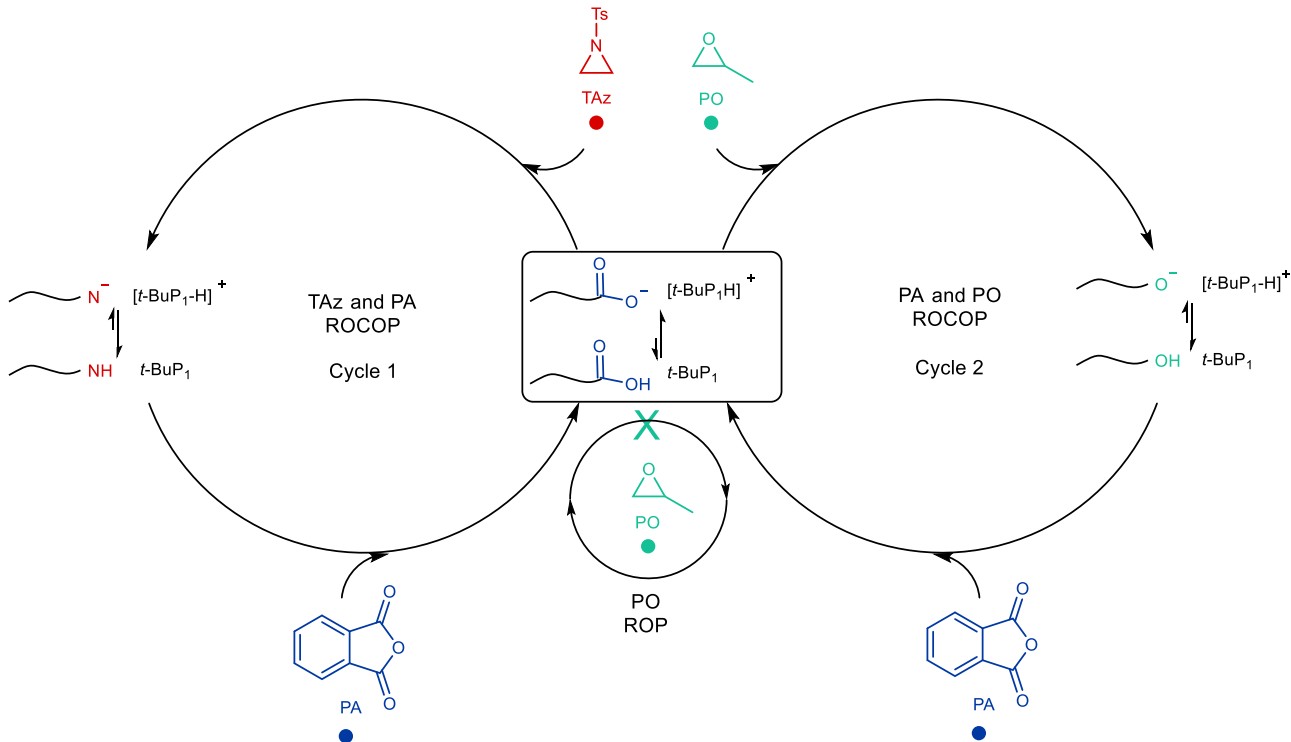

**Fig. 7 Proposed terpolymerization pathways.** Chemoselective TAz/PA ROCOP and PA/PO ROCOP by a simple organocatalyst, $t$-BuP$_1$.

complex corresponds to dormant species that can suppress the activation of $t$-BuP$_1$-H$^+$ to PO (Supplementary Fig. 53). After the complete consumption of TAz, the $t$-BuP$_1$-H$^+$ released from $t$-BuP$_1$ switches to activated species and starts the second copolymerization cycle (PA/PO ROCOP). Similar proton shuttling between the catalysts and growing chain end is considered key to promoting selectivity and control for the copolymerization[45]. For both copolymerization cycles, the phosphazenium shows high monomer selectivity and polymerization controllability.

The reactivity ratios were determined by real-time $^1$H NMR kinetics using the Fineman–Ross and Mayo–Lewis methods. The alternating characteristic is observed for TAz/PA copolymerization cycle ($r_{TAz} = 0.0089$ and $r_{PA} = 0.0052$, Supplementary Figs. 68 and 69), showing that the copolymerization rate of TAz/PA is faster than the homopolymerization rate of TAz. The PA/PO copolymerization shows a perfect alternating characteristic ($r_{PA} = r_{PO} = 0$) because the PA and PO cannot be homopolymerized by $t$-BuP$_1$ at 100 °C for 5 h. The extreme reactivity difference of TAz/PO indicates that the homopolymerization rate of TAz is faster than the copolymerization rate of TAz/PO[48]. All the reactivity ratios support the following: copolymerization rate of TAz/PA > homopolymerization rate of TAz ≫ copolymerization rate of TAz/PO > homopolymerization rate of PA or homopolymerization rate of PO.

**Extension to other monomers, initiators, and catalysts**. Other epoxides were used in the terpolymerization to synthesize diverse diblock dialternating terpolymers by a "one-pot/one-step" procedure. Ethylene oxide (EO) with higher monomer reactivity than PO accelerates the second-stage copolymerization rate (entries 9–11 in Table 1 and Supplementary Table 5). The $^1$H NMR, $^{13}$C{$^1$H} NMR spectra, SEC traces, and DOSY spectrum (Supplementary Figs. 8–12) confirm the successful synthesis of diblock dialternating terpolymer poly(TAz-*alt*-PA)-*b*-poly(PA-*alt*-EO). A little transesterification reaction is observed at high PA conversion at 100 °C because of the high nucleophilic character of chain-

end alkoxide from EO ring opening. But the transesterification reaction can be successfully avoided by decreasing the reaction temperature to 60 °C (Supplementary Fig. 9). Butylene oxide (BO) with similar reactivity with PO can be used too in the terpolymerization with TAz and PA, to synthesize diblock dialternating terpolymer (Supplementary Figs. 13–16). Styrene oxide (SO), 1,2-epoxy-3-phenoxypropane, $n$-buty glycidyl ether, and CHO were evaluated in the terpolymerization with TAz and PA (Supplementary Figs. 26–29). Incomplete diblock dialternating terpolymers [a mixture of poly(TAz-*alt*-PA)-*b*-poly(PA-*alt*-epoxides) and poly(TAz-*alt*-PA)] and a decrease in the alternating degree for TAz/PA ROCOP were observed by $^1$H NMR and SEC. We attempted to prepare perfect diblock dialternating terpolymers by changing temperature (60 °C), solvent (THF and DMF), and amount of $t$-BuP$_1$ ($t$-BuP$_1$:In = 2:1) without success. The polarity of the epoxides, including dielectric constant and dipole moment, may be the reason for the slow or incomplete dissociation of the chain end in the second copolymerization cycles, resulting in inefficient initiation and the formation of incomplete diblock dialternating terpolymers. The substitution groups on the epoxides may hinder the nucleophilic attack of the chain end and lead to incomplete initiation. Two more $N$-sulfonyl aziridines were synthesized and used in the terpolymerization with PA and PO. The high active $N$-brosylaziridine (BAz) and $N$-(4-nitrobenzenesulfonyl) aziridine (NAz) accelerate the first copolymerization stage (Supplementary Tables 14 and 15). The $^1$H NMR, $^{13}$C{$^1$H} NMR spectra, SEC traces, and DOSY spectrum (Supplementary Figs. 30–36) confirm the successful synthesis of diblock dialternating terpolymers. Bifunctional initiators, 1,4-benzenedimethanol and polyethylene glycol ($M_n = 8000$, PEG8000), were used to synthesize ABCBA-type pentablock terpolymers (entries 18–21 in Table 1). We did not observe any methylene signal peak of the 1,4-benzenedimethanol (4.73 ppm, Supplementary Fig. 19) on the terpolymers (Supplementary Fig. 18), indicating that both hydroxyl groups on the 1,4-benzenedimethanol can initiate the terpolymerization. Well-controlled

diblock alternating terpolymers with narrow molecular weight distributions are obtained in the absence of a tapered sequence (Supplementary Figs. 17–24).

Other catalysts ($t$-BuP$_4$, $t$-BuP$_2$, DBU, and TBD) were evaluated in the terpolymerization (entries 22–27 in Table 1, Supplementary Figs. 37–40, and Supplementary Tables 16 and 17). $t$-BuP$_4$ and $t$-BuP$_2$ possessing higher basicity than $t$-BuP$_1$ accelerate the first copolymerization cycle of TAz/PA and remain the controlled polymerization behavior (entries 22–25 in Table 1, Supplementary Tables 16 and 17, and Supplementary Figs. 37 and 38). However, the SEC traces show broad molecular weight distributions ($Đ$: 1.13–1.77) in the second copolymerization of PA/PO, indicating the presence of extensive transesterification. In the case of terpolymerization catalyzed by DBU and TBD (entries 26–27 in Table 1 and Supplementary Figs. 39 and 40), the ether (3.30–3.60 p.p.m.) and amine linkage (3.20–3.30 p.p.m.) were observed by NMR in both copolymerizations, indicating poor alternating characteristics.

Initial monomers feed ratios (TAz/PA) were changed to 1/2 and 1/4 in the terpolymerization (entries 28–31 in Table 1 and Supplementary Figs. 41 and 42). The $^1$H NMR and SEC traces (Supplementary Figs. 41 and 42) confirm the successful synthesis of diblock dialternating terpolymers.

**Thermal analysis**. The diblock dialternating terpolymers were analyzed by differential scanning calorimetry (DSC, Supplementary Figs. 64 and 65) and thermogravimetric analysis (Supplementary Figs. 66 and 67). Poly(TAz-$alt$-PA)$_{30}$-$b$-poly(PA-$alt$-PO)$_{60}$ shows a single glass transition temperature ($T_g$) of 72 °C between the $T_g$ of poly(TAz-$alt$-PA)$_{30}$ (114 °C) and poly(PA-$alt$-PO)$_{60}$ (59 °C), indicating a complete mixing of the two alternating blocks, due to the common PA monomer. Decreasing the polymerization degrees of second block increases the $T_g$ value [$T_g$ of Poly(TAz-$alt$-PA)$_{30}$-$b$-poly(PA-$alt$-PO)$_{30}$ is 78 °C], and vice versa [$T_g$ of Poly(TAz-$alt$-PA)$_{30}$-$b$-poly(PA-$alt$-PO)$_{90}$ is 64 °C]. Diblock dialternating terpolymers synthesized by other organic base catalysts($t$-BuP$_4$, $t$-BuP$_2$, DBU, and TBD) showed lower $T_g$ values (61–68 °C, Supplementary Fig. 65). Compared with homopolymer poly(PA-$alt$-PO)$_{60}$ ($T_{d\ 5\%}$ = 327 °C), all the diblock dialternating terpolymers show lower thermal stability ($T_{d\ 5\%}$ = 251–266 °C) (Supplementary Figs. 66 and 67).

In summary, diblock terpolymers with two alternating sequences, poly($N$-sulfonyl aziridine-$alt$-phthalic anhydride)-$b$-poly(phthalic anhydride-$alt$-epoxide), were synthesized by terpolymerization of $N$-sulfonyl aziridines, PA, and epoxides without additional actions ("one-pot/one-step"). Two alternating copolymerization cycles have been proposed, which include ROCOP of $N$-sulfonyl aziridines/PA and ROCOP of PA/epoxides. The proposed organocatalyst $t$-BuP$_1$ auto-switches between two different alternating copolymerization cycles resulting in high monomer selectivity and kinetic controllability. The obtained diblock dialternating terpolymers have a sharp junction between the two block segments rather than tapered sequences. Terpolymers with two precise alternating structures, controllable molecular weights, and narrow molecular weight distributions ($Đ$ < 1.08) were obtained through the terpolymerization process. The synthetic strategy of diblock terpolymers with two alternating sequences in a "one-pot/one-step" procedure is promising and profound for industrial development. Furthermore, with the use of multifunctional initiators, as in the case of the synthesized pentablock, this methodology opens new horizons to dialternating-based complex macromolecular architectures.

## Methods

**Materials**. $t$-BuP$_1$ was supplied by Aldrich Chemicals and used without further purification. TAz (98%), 1,4-benzenedimethanol, and PEG8000 were supplied by

Aldrich Chemicals and were dried over phosphorus pentoxide overnight twice. PA (Aldrich ≥99%) was provided by Aldrich Chemicals and was purified by heating a 10 wt.% solution in CHCl$_3$ under reflux for 30 min, followed by hot filtration through Celite. The filtrated PA was recrystallized at room temperature, and the obtained PA crystal was sublimated at 100 °C twice under a dynamic vacuum. Then, the PA was freeze-dried in dry 1,4-dioxane, followed by removal of water traces under vacuum in the presence of phosphorus pentoxide. 1,4-Dioxane, PO, and BO were dried over calcium hydride overnight, distilled, and then dried over $n$-butyllithium for 2 h, distilled, and then stored in a glovebox before use. EO was dried over sodium flakes overnight, distilled, and then stored in the refrigerator inside the glovebox. BAz and NAz were synthesized by ethanolamine and corresponding sulfonyl chloride in Supplementary Methods.

**General procedure for the terpolymerization of $N$-sulfonyl aziridine, PA, and epoxide**. As an example, the procedure corresponding to entry 7 (Table 1) is given below. Eleven 100 mL dry Schlenk thick reactors were prepared and charged, in an argon-filled glovebox, with the same feed ratio, which is PA (0.264 g, 1.8 mmol), TAz (0.116 g, 0.6 mmol), initiator ($N$-benzyl-4-methylbenzenesulfonamide) (0.0052 g, 0.02 mmol), and anhydrous epoxides (0.6 mL) (ideal gas law was used to estimate internal pressure). After stirring for 5 min, $t$-BuP$_1$ (4.6 μL, 0.02 mmol) was added to each mixture. The reactors were sealed and then removed from the glovebox and stirred at 100 °C to initiate the polymerization. At a specific time, a reactor was taken out of the oil bath and quickly cooled down in an ice-water bath. A sample was taken out from the crude product for SEC and NMR analysis. The produced terpolymer was precipitated by the slow addition of the diluted reaction solution (CHCl$_3$) to an excess of methanol. The precipitation process was repeated three times, and the terpolymers were dried in a vacuum oven at 40 °C overnight. Isolated yield = 0.40 g, 87% (before taking samples). The procedure was repeated three times and the average values, as well the standard error bars, are given in the graphs.

Another example of the procedure corresponding to entry 11 (Table 1) is the following. Eleven 100 mL dry Schlenk thick reactors were prepared and charged, in an argon-filled glovebox, with the same feed ratio, which is PA (0.264 g, 1.8 mmol), TAz (0.116 g, 0.6 mmol), and initiator ($N$-benzyl-4-methylbenzenesulfonamide) (0.0052 g, 0.02 mmol) (ideal gas law is used to estimate internal pressure). The reactors were placed in a freezer inside a glovebox (−20 °C) for 60 min. The reactors were removed from the freezer, and anhydrous EO (0.6 mL) was added to each reactor. After stirring for 5 min, the reactors were placed back in the freezer (−20 °C) for 60 min, and then $t$-BuP$_1$ (4.6 μL, 0.02 mmol) was added to each cold mixture. The reactors were sealed and then taken out of the glovebox and stirred at 100 °C to initiate the polymerization. At a specific time, a reactor was taken out of the oil bath and quickly cooled down in an ice-NaCl bath. A sample was taken from the crude product for SEC and NMR analysis. The produced terpolymer was precipitated by the slow addition of the diluted reaction solution (CHCl$_3$) to an excess of methanol. The precipitation process was repeated three times, and the terpolymers were dried in a vacuum oven at 40 °C overnight. Isolated yield = 0.39 g, 89% (before taking samples). The procedure was repeated three times, and the average value, as well the standard error bars, are given in the graphs.

In situ FTIR study was conducted using a ReactIR 45m (Mettler Toledo) and equipped with a dip probe. The background was collected after an empty Schlenk flask was flame-dried under a vacuum. The probe was transferred into the argon-filled glovebox. The tip of the probe was immersed in the reaction mixture (the mixture prepared by the entry 7 procedure). The reactors were sealed and then taken out of the glovebox and stirred at 100 °C to start the polymerization. The spectra were collected every 1 min, and each spectrum was scanned 256 times.

**Characterization**. Nuclear magnetic resonance ($^1$H NMR and $^{13}$C{$^1$H} NMR) measurements were recorded on a Bruker AVANCE III-400 Hz instrument in CDCl$_3$ at room temperature. DOSY measurements were recorded on a Bruker AVANCE III-600 Hz instrument in CDCl$_3$ at room temperature. SEC measurements were performed at 35 °C with THF as an eluent at a flow rate of 1.0 mL min$^{-1}$, with a Viscotek GPC$_{max}$ VE2001 System, and PSS columns (Styragel HR 3, 4, and 5). The molecular weight distributions ($M_w/M_n$, $Đ$) were determined by conventional SEC analysis using a calibration curve from polystyrene standards. MALDI-ToF mass measurements were recorded in linear mode, using 2,5-dihydroxybenzoic acid as the matrix in THF (sample/matrix: 1/50). In situ FTIR study was conducted using a ReactIR 45m (Mettler Toledo). The samples were collected every 1 min, and each spectrum was scanned 256 times. DSC measurements were performed at a heating rate of 10 °C min$^{-1}$ on a Mettler Toledo DSC1/TC100 system under a nitrogen atmosphere. The curve of the second heating scan was adopted to determine the $T_g$. NMR titration experiments were performed in Young's NMR tubes using anhydrous toluene-$d_8$ solvent under argon.

## Data availability

The authors declare that the data supporting this study are available within the paper and the Supplementary information File. All other data is available from the authors upon request.

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

## Acknowledgements

The research work was supported by King Abdullah University of Science and Technology (KAUST), Thuwal, Saudi Arabia.

## Author contributions

J.X. and N.H. conceived the idea and designed the experiments. J.X. performed the experiments. J.X. and X.W. performed the mechanism verification. All authors analyzed the results and co-wrote the manuscript.

## Competing interests

The authors declare no competing interest.
