## [Peer Review File · Nature Communications]

Reviewers' comments:

Reviewer #1 (Remarks to the Author):

Data presentation, referencing and methodology are fine. It is solid work that should be published but its novelty is not very high considering the very ample literature that exists in the field. Although it is of interest to some part of the chemical science community, its wider impact (and therefore its relevance to Nature Communications) is questionable.

Indeed, the authors state that their work shows how to use their catalyst (tert-butylimino-tris(dimethylamino)phosphorene) with three different monomers to produce diblock dialternating copolymers in a one-pot sequence; the same principles should apply, in the future, to many other epoxides, anhydrides, aziridines and heterocycles.

Although these claims are novel (but somewhat overstated), I do not believe that they represent a significant advance in the field. Indeed, from a reactivity point of view, the authors have already published the same catalytic system for the ring-opening copolymerization of N-sulfonyl aziridines and cyclic anhydrides (ref 35). The one-pot synthesis of these copolymers is however interesting and should make it possible to obtain interesting polymers. Unfortunately, the polymers obtained, even if they are new, still seem to me very limited in terms of structural diversity. Therefore, examples of more varied monomers are still missing in this paper. This would perhaps allow to obtain more significant molecular weights and to demonstrate the interest of modulating the structure of complexes to vary the physical properties of materials.

In order to reinforce the conclusions of this work, it is to me essential to show the impact of the catalyst design on the properties of the polymers. Indeed, the precision placement of particular monomer sequences or blocks is expected to allow control over macroscopic properties, such as glass transition temperature, viscosity or modulus. This is a decisive point to show the interest of the formed macromolecules.

As it stands, I find that this work would be better suited to a more specialized journal (in polymer chemistry) because novelty in terms of catalysis is too limited.

Reviewer #2 (Remarks to the Author):

This manuscript by Hadjichristidis and colleagues describes the selective synthesis of diblock copolymers, where each block is an alternating copolymer. The results are very interesting, as the preparation of diblock dialternating copolymers from a one-pot, one-step method is a significant synthetic challenge. Aziridine/anhydride ring-opening copolymerisation remains relatively underexplored, and I believe the formation of block copolymers from this route will be of significant interest to the community. The selectivity of this route appears to be quite remarkable, as even when performed in neat epoxide, epoxide/anhydride ring-opening copolymerisation does not occur until all of the aziridine is consumed. The addition of further aziridine then switches the reaction back to the aziridine/anhydride copolymerisation, demonstrating the potential to form multiblock copolymers from this route. I believe this article will be of significant interest to the readership of Nature Communications and would recommend this work for publication providing the following points are addressed.

The proposed mechanism:

1) After all the aziridine has been consumed, it would be interesting to know why the first epoxide is very slow to open (i.e. why there is a “dormant period” between the aziridine/anhydride and epoxide/anhydride polymerisations). This delay a little surprising – it seems that the first equivalent of epoxide is slow to ring-open (from an opened anhydride chain end), yet subsequent epoxide insertions (into the opened anhydride chain ends) are much faster. Can the authors provide any insight into this observation?

2) Figure S36 is very interesting. Could the significant shift observed for the organocatalyst (tBuP1) in the absence/presence of the aziridine be because the organocatalyst preferentially interacts with the aziridine instead of the epoxide as shown (top), which may activate the aziridine towards ring-opening? Is there any possibility that proton transfer from the initiator to the aziridine could occur (and if so, enhance ring-opening), instead of proton transfer to the organocatalyst? I would add a ^1H NMR spectra of the combination of initiator and aziridine, to see whether any proton transfer could be observed. It may also be helpful to include the ^1H NMR of the catalyst and initiator as separate species, to confirm the proposed proton transfer between these two species. Note that in this figure, there are currently two spectra labelled as b.

3) The proposed catalytic cycle shown in Figure 7 needs some attention. The aziridine and epoxide (top left and top right) should be shown as entering into the catalytic cycles, rather than as intermediates formed from the active opened anhydride/organocatalyst species (centre). The charges do not balance as shown (e.g. should the organocatalyst have a proton on the left hand side structure with the ring-opened aziridine)? The bottom figures should also include the organocatalyst to balance the equations. I would also recommend removing “*” from the bottom figures (or explaining what this symbol refers to), as “*” is sometimes used to denote an active chain end in the polymer literature.

Additional points:

4) "The benzene signals of TAz" - change benzene for aromatic.

5) I would recommend rephrasing "The molecular weight of the copolymer increases with the reaction time, in agreement with the theoretical molecular weight of poly(TAz-alt-PA) (Figure 6b)." This statement may be misleading; the copolymer MW cannot be directly compared to the theoretical weight, as the SEC values were determined by calibration against polystyrene standards.

6) "The decreased temperature slows down the reaction rates of both the first-stage (kobs, first-stage = 11.2547 M-1h-1 in 100o C; kobs, first-stage = 0.4734 M-1h-1 in 60o C) and the second-stage copolymerization (kobs, second-stage = 0.4500 M-1h-1 in 100o C; kobs, second-stage = 0.0682 M-1h-1 in 60o C) (Figure S31-S34)." It may be helpful to show how these numbers were derived from Figures S31-34. Should one of these figures also show the second stage copolymerisation at 60oC?

7) "All the reactivity ratios support the following: copolymerization rate of TAz/PA > homopolymerization rate of TAz > copolymerization rate of TAz/PO > homopolymerization rate of PA/PO." – should homopolymerization rate of PA/PO be copolymerisation instead?

8) "Furthermore, with the use of multifunctional initiators, as in the case of the synthesized pentablock, this methodology opens new horizons to dialternating-based complex macromolecular architectures with unprecedented properties." It would be helpful to include evidence to support the pentablock structure (or to make it clear that this assignment is somewhat speculative), as it is currently unclear whether initiation occurs from one or both ends of the diols.

ESI

9) The experimental states "1,4-Dioxane, propylene oxide (PO), and butylene oxide (BO) were dried over calcium hydride, over n-butyllithium overnight, distilled, and stored in a glovebox before use." Does n-butyllithium not react with (ring-open) the epoxides (this has been documented in literature)?

10) The reactions were typically performed at boiling points much higher than the boiling points of the epoxides. It would be important to acknowledge the pressure build-up (and associated safety aspects) of performing the reactions under these conditions. It would also be important to explain how the reaction was set up when using ethylene oxide, which is a gas at room temperature.

11) Table S10 footnote e, it would be helpful to clarify when TAz was added.

12) Fig S31 – It would help to add a few extra data points if possible.

Reviewer #3 (Remarks to the Author):

The results in this manuscript represent a continuous work on the ring-opening copolymerization (ROCOP) of aziridines and cyclic anhydrides (Angew. Chem. Int. Ed. 2021, 60, 6949–6954), in which the authors extended the process to a “one pot/one step” terpolymerization employing aziridines, cyclic anhydrides and epoxides. Interestingly, the authors found that the copolymerization of aziridines and cyclic anhydrides is thermodynamically favored with respect to epoxide/anhydride copolymerization, leading to di- or tri-blocks copolymers (when using a bifunctional initiator) with perfectly alternating segments of aziridine/anhydride and epoxide/anhydride, thus affording poly(ester amides)-b-poly(ester) block copolymers. Impressively, the latter are formed without any incorporation of epoxide into the aziridine/anhydride segments despite performing the terpolymerization in neat epoxide. The authors provide compelling evidences for perfect selectivity in monomer enchainment, using NMR, IR, MALDI-ToF and SEC techniques, and supported by DFT calculations of thermodynamic parameters. Furthermore, the authors show that by feeding an additional portion of aziridine to the reaction mixture during the second ROCOP reaction (epoxide/anhydride), the latter is inhibited and a shift to the first ROCOP cycle occurs until complete consumption of the aziridine monomer. Such process has therefore the potential to lead to tailor-made block copolymers of alternating poly(ester amides) and poly(ester) segments. The NMR results are comprehensive and validate the sequence of ROCOP reactions, including a dormant stage between the two phases. The IR and SEC data complement the NMR results, further corroborating the selectivity in monomer incorporation and the living nature of the catalytic system. Nevertheless, the authors do not provide any physical properties of the polymers obtained (with the exception of a single Tg value) which makes it hard to evaluate whether these polymers have valuable industrially-relevant potential, as stated in the introduction and conclusions. In addition, there are a few issues that the authors should address:

- 1) In the context of switchable “one pot/one step” polymerization the work by Williams and coworkers should be cited (Catal. Sci. Technol., 2021,11, 1737-1745; ; Chem. Commun., 2019, 55, 7315).
- 2) Page 4: the authors should refer to the initial monomers feed ratio when discussing the % conversions in order to avoid confusion. In this context, the authors did not explain why a 1:3 Taz/PA ratio was chosen (instead of 1:2 for example).
- 3) Page 9: the NMR shifts ascribed to the poly(PA-alt-PO) segment are methine and methylene signals, and not “methylene and methyl group” as noted for j and k in Figure 4a.
- 4) Page 9: “There are no ether linkage signals” instead of “There is no ether linkages signal”, as these would be two signals.
- 5) Page 9: the authors state that “The characteristic signal peak of the carbonyl group from poly(PA-alt-PO) was observed at 167.0 ppm”, however there are two carbonyl signals in the 13C NMR spectrum ascribed to both carbonyl groups of PA which are de-symmetrized upon copolymerization with PO.
- 6) Page 10: Figure 4a is too small which makes extremely hard to follow the signal annotations.

7) Page 13: the catalytic cycles are drawn wrong; the monomers should not be part of the cycle but rather enter the cycle by external arrows. Further, the amine anion (and corresponding alkoxide in the cycle 2) should react with PA (monomer B) prior to forming a polymeryl chain with a carboxylate end group, in contrast to the drawing in which it looks as if a polymer (without any reactive end group) is reacting with the monomer. Finally, the same overall charge should be maintained throughout the cycle (the catalytic base should balance the reactive anionic end groups as in the conjugation of the two cycles).

8) Page 16: the authors state that the steric hindrance of the bulkier epoxides is responsible for the incomplete terpolymerization and a decrease in the alternating nature of Taz/PA. I would expect a bulkier epoxide to increase the selectivity in Taz/PA insertion given the slower rate of ring-opening observed by the authors. In addition, based on the NMR figure it seems that PA was completely converted so the polymer mixture obtained might be the consequence of inefficient initiation of the second ROCOP process. The authors did not provide any SEC data (M_n vs. theoretical M_n , PDI) that will allow to hypothesize about this issue.

9) SI, page 2: Overall, the experimental data is well presented, however the authors neglected to explain how exactly they acquired the conversion/kinetic data. The reactions were performed at a high temperature (100 °C), which exceeds the boiling point of THF and most employed epoxides. Thus, the authors used a Schlenk reactor, which I presume could support the internal pressure formed during the reaction, and performed the reaction outside the glove box. Therefore, the authors should comment on how the data for the polymerization was acquired. Did they withdraw aliquots from a single reaction or did they performed several processes simultaneously, quenching the each of the reactions at different reaction times? If the latter was performed, did the reactions were performed in triplicates? Are the presented data points average values (in which case an error should be reported as well)?

10) SI: Did the authors use an internal standard for the kinetic measurements in order to determine the accuracy of the concentrations and resulting conversions? If they did, it should be mentioned in the SI section.

11) SI: The in-situ FTIR instrument does not appear in the instrumentation section of the SI and the authors did not comment on the experimental details of its usage. Did the authors use an instrument equipped with an internal probe that could be inserted into the reaction mixture? The authors should elaborate on the experimental setup and data acquisition methods.

12) SI, page 25: condition "b" appears twice in Figure S36. In addition, this NMR evidence for the inhibition of the propagating carboxylate by residual TAz is not compelling enough. Upon the addition of TAz the chemical shift of the carboxylate's alpha proton is only slightly affected, thus rendering its supposed shielding doubtful. In addition, similar interactions could be envisioned in the copolymerization of EO a between the carboxylate and the methylene protons of the epoxide (which is in a large excess), which would lead to epoxide ring-opening during a polymerization process. However, a similar dormant stage appears in the ROCOP of EO as well. The authors should consider to revisit this experiment, possibly performing several control experiments in which the carboxylate will be treated separately with TAz and EO, followed by titration of each reaction with the opposed monomer.

Comments from the reviewers and our responses:

Replies to Reviewer(s)' Comments

Reviewers' comments:

Reviewer #1 (Remarks to the Author):

Data presentation, referencing and methodology are fine. It is solid work that should be published but its novelty is not very high considering the very ample literature that exists in the field. Although it is of interest to some part of the chemical science community, its wider impact (and therefore its relevance to Nature Communications) is questionable.

Answer: We thank the Reviewer for his critical that made our work better.

We believe that our work is highly original, and as far as we know, there is no other paper dealing with similar work. The explanation is given below. We synthesized pure diblock dialternating copolymers (ABABAB-*block*-CBCBC) without any tapered sequences between the two blocks from a mixture of three monomers without additional actions. This was very challenging, however we were able to design for the first time the monomers (reactivity) and catalyst (selectivity). To our knowledge, only one terpolymerization system (epoxide, anhydrides, and CO₂) was reported until now but with poor monomer selectivity, which led to a tapered chain (over 10%) between the block, as the authors admitted (Refs 39-43 in the MS). It is well known that the tapered chain between the two neat blocks affect their properties (Ref. 44 in the MS). We believe that our work has a wide impact since it provides a universal verification method to distinguish between diblock copolymers having a tapered chain at the junction point from pure diblock copolymers.

Indeed, the authors state that their work shows how to use their catalyst (tert-butylimino-tris(dimethylamino)phosphorene) with three different monomers to produce diblock dialternating copolymers in a one-pot sequence; the same principles should apply, in the future, to many other epoxides, anhydrides, aziridines and heterocycles.

Answer: We believe that this work will inspire others to use our principle to terpolymerization of the monomer given by the Reviewer as well as other monomers. In addition, the proposed verification method to distinguish between the tapered block copolymer and pure block copolymer is universal and can be applied for many other monomers.

Although these claims are novel (but somewhat overstated), I do not believe that they represent a significant advance in the field. Indeed, from a reactivity point of view, the authors have already published the same catalytic system for the ring-opening copolymerization of N-sulfonyl aziridines and cyclic anhydrides (ref 35). The one-pot synthesis of these copolymers is

however interesting and should make it possible to obtain interesting polymers. Unfortunately, the polymers obtained, even if they are new, still seem to me very limited in terms of structural diversity. Therefore, examples of more varied monomers are still missing in this paper. This would perhaps allow to obtain more significant molecular weights and to demonstrate the interest of modulating the structure of complexes to vary the physical properties of materials.

Answer: As we explained above the claims are novel and not overstated. Moreover, monomers (three instead of two), catalyst (*t*-BuP₁ instead of *t*-BuP₄) and polymerization mechanism (self-switchable instead of anionic) is completely different than in our previous work (ref. 37). This it was very challenging to design the polymerization system.

We have already demonstrated that our claim is valid for other monomers (different epoxides) and other initiators (BnN(H)Ts, 1,4-benzenedimethanol and polyethylene glycol).

Following the Reviewer's comment, we synthesized many dialternating diblock with different molecular weights and we studied the thermal properties of the synthesized terpolymers by DSC and TGA. The results given below and in SI.

Initial monomer feed ratios (Table S16-S17, and Figure S34-S35) were changed to obtain the different molecular weights, and some thermodynamics physical properties were added (Figure S57-S60).

8. Terpolymerization of TAz, PA, and PO with different monomer feed ratio

Table S16. Terpolymerizations of TAz, PA and PO using *t*-BuP₁^a

Entry	Time	Conv. (TAz) ^b /%	Conv. (PA) ^b /%	$M_{n,theo}$ ^c /kg mol ⁻¹	$M_{n,NMR}$ ^b /kg mol ⁻¹	\mathcal{D} ^d
1	2 min	25	12	2.85	2.96	1.07
2	5 min	63	21	6.79	6.98	1.04
3	10 min	94	47	10.0	10.8	1.03
4	15 min	99	49	10.5	11.2	1.03
5	30 min	99	49	10.5	11.2	1.03
6	1 h	99	52	10.9	11.9	1.06
7	2 h	99	69	13.0	13.7	1.04
8	3 h	99	88	15.3	15.9	1.04
9	4h	99	99	16.7	18.2	1.04

^aThe terpolymerizations were performed at a ratio of [TAz]₀/[PA]₀/[BnN(H)Ts]₀/[*t*-BuP₁]₀ = 30/60/1/1 in PO ([TAz]₀ = 1.0 M) at 100°C. ^bDetermined by ¹H NMR in CDCl₃ using integrals of the characteristic signals. ^cCalculated as follows: (M.W. of Initiator) + ([TAz]₀/[I]₀) × conv.(TAz) × (M.W. of TAz + M.W. of PA) + {([PA]₀/[I]₀) × conv.(PA) - ([TAz]₀/[I]₀) × conv.(TAz)} × (M.W. of epoxides + M.W. of PA). ^dDetermined by SEC at 35°C in THF (1.0 mL min⁻¹) using Pst standards.

Figure S34. Stacked ^1H NMR spectra (400 MHz, CDCl_3 , 25°C) and SEC traces (THF, 35°C) of the reaction mixture at the ratio of $[\text{Taz}]_0/[\text{PA}]_0/[\text{BnN}(\text{H})\text{Ts}]_0/[t\text{-BuP}_1]_0 = 30/60/1/1$ in PO ($[\text{Taz}]_0 = 1.0 \text{ M}$) at 100°C (entries 28 and 29, Table 1).

Table S17. Terpolymerizations of TAz, PA and PO using $t\text{-BuP}_1$ ^a

Entry	Time	Conv. (TAz) ^b /%	Conv. (PA) ^b /%	$M_{n,\text{theo}}^c/\text{kg mol}^{-1}$	$M_{n,\text{NMR}}^b/\text{kg mol}^{-1}$	\bar{D}^d
1	2 min	36	9	4.00	4.31	1.07
2	5 min	77	19	8.24	8.98	1.05
3	10 min	99	24	10.5	11.2	1.03
4	15 min	99	24	10.5	11.2	1.03
5	30 min	99	24	10.5	11.2	1.03
6	1 h	99	30	11.8	13.1	1.06
7	2 h	99	35	13.1	14.2	1.04
8	3 h	99	47	16.0	16.9	1.04
9	4 h	99	58	18.8	20.1	1.04
10	6 h	99	88	26.2	27.8	1.03
11	8 h	99	99	28.9	30.0	1.02

^aThe terpolymerizations were performed at a ratio of $[\text{Taz}]_0/[\text{PA}]_0/[\text{BnN}(\text{H})\text{Ts}]_0/[t\text{-BuP}_1]_0 = 30/120/1/1$ in PO ($[\text{Taz}]_0 = 1.0 \text{ M}$) at 100°C . ^bDetermined by ^1H NMR in CDCl_3 using integrals of the characteristic signals. ^cCalculated as follows: (M.W. of Initiator) + $([\text{Taz}]_0/[\text{I}]_0) \times \text{conv.}(\text{TAz}) \times (\text{M.W. of TAz} + \text{M.W. of PA}) + \{([\text{PA}]_0/[\text{I}]_0) \times \text{conv.}(\text{PA}) - ([\text{Taz}]_0/[\text{I}]_0) \times \text{conv.}(\text{TAz})\} \times (\text{M.W. of epoxides} + \text{M.W. of PA})$. ^dDetermined by SEC at 35°C in THF (1.0 mL min^{-1}) using PSt standards.

Figure S35. Stacked ¹H NMR spectra (400 MHz, CDCl₃, 25°C) and SEC traces (THF, 35°C) of the reaction mixture at the ratio of [TAz]₀/[PA]₀/[BnN(H)Ts]₀/[*t*-BuP₁]₀ = 30/120/1/1 in PO ([TAz]₀ = 1.0 M) at 100°C (entries 30 and 31, Table 1).

12. Thermodynamics analysis

Figure S57. DSC thermogram for poly(TAz-*alt*-PA)₃₀, poly(PA-*alt*-PO)₆₀, poly(TAz-*alt*-PA)₃₀-*b*-poly(PA-*alt*-PO)₆₀, poly(TAz-*alt*-PA)₃₀-*b*-poly(PA-*alt*-EO)₆₀, and poly(TAz-*alt*-PA)₃₀-*b*-poly(PA-*alt*-BO)₆₀ under N₂.

Figure S58. DSC thermogram for poly(TAz-*alt*-PA)₃₀-*b*-poly(PA-*alt*-PO)₃₀, poly(TAz-*alt*-PA)₃₀-*b*-poly(PA-*alt*-PO)₉₀, poly(TAz-*alt*-PA)₃₀-*b*-poly(PA-*alt*-PO)₆₀ by *t*-BuP₄, poly(TAz-*alt*-PA)₃₀-*b*-poly(PA-*alt*-PO)₆₀ by *t*-BuP₂, poly(TAz-*alt*-PA)₃₀-*b*-poly(PA-*alt*-PO)₆₀ by DBU, and poly(TAz-*alt*-PA)₃₀-*b*-poly(PA-*alt*-PO)₆₀ by TBD.

Figure S59. TGA thermogram for poly(TAz-*alt*-PA)₃₀, poly(PA-*alt*-PO)₆₀, poly(TAz-*alt*-PA)₃₀-*b*-poly(PA-*alt*-PO)₆₀, poly(TAz-*alt*-PA)₃₀-*b*-poly(PA-*alt*-EO)₆₀, and poly(TAz-*alt*-PA)₃₀-*b*-poly(PA-*alt*-BO)₆₀.

Figure S60. TGA thermogram for poly(TAz-*alt*-PA)₃₀-*b*-poly(PA-*alt*-PO)₃₀, poly(TAz-*alt*-PA)₃₀-*b*-poly(PA-*alt*-PO)₉₀, poly(TAz-*alt*-PA)₃₀-*b*-poly(PA-*alt*-PO)₆₀ by *t*-BuP₄, poly(TAz-*alt*-PA)₃₀-*b*-poly(PA-*alt*-PO)₆₀ by *t*-BuP₂, poly(TAz-*alt*-PA)₃₀-*b*-poly(PA-*alt*-PO)₆₀ by DBU, and poly(TAz-*alt*-PA)₃₀-*b*-poly(PA-*alt*-PO)₆₀ by TBD.

In order to reinforce the conclusions of this work, it is to me essential to show the impact of the catalyst design on the properties of the polymers. Indeed, the precision placement of particular monomer sequences or blocks is expected to allow control over macroscopic properties, such as glass transition temperature, viscosity or modulus. This is a decisive point to show the interest of the formed macromolecules. As it stands, I find that this work would be better suited to a more specialized journal (in polymer chemistry) because novelty in terms of catalysis is too limited.

Answer: We added the following paragraph in order to clarify the topic according to the Reviewer's comment.

Other catalysts (*t*-BuP₂, *t*-BuP₄, DBU, and TBD) were evaluated in the terpolymerization (entries 22-27 in Table 1, Figure S30-S33, and Table S14-S15). *t*-BuP₄ and *t*-BuP₂ possessing higher basicity than *t*-BuP₁ accelerate the first copolymerization cycle of TAz/PA and remain the controlled polymerization behavior (entries 22-25 in Table 1, Table S14-S15, and Figure S30-S31). However, the SEC traces show broad molecular weight distribution (\bar{D} : 1.13 to 1.77) in the second copolymerization of PA/PO, indicating the presence of extensive transesterification. In the case of terpolymerization catalyzed by DBU and TBD (entries 26-27 in Table 1 and Figure S32-33), the ether (3.30-3.60 ppm) and amine linkages (3.20-3.30 ppm) were observed by NMR in both copolymerizations, indicating poor alternating characteristics.

The results are given below and in SI.

Concerning Reviewer's comment about suitability to Polymer Chemistry Journal we think he will be convinced by now that it is better suitable for Nature Communications.

7. Terpolymerization of TAz, PA, and PO with different bases catalysts

Table S14. Terpolymerizations of TAz, PA and PO using *t*-BuP₄^a

Entry	Time	Conv. (TAz) ^b /%	Conv.(PA) ^b /%	$M_{n,theo}^c$ /kg mol ⁻¹	$M_{n,NMR}^b$ /kg mol ⁻¹	\bar{D}^d
1	2 min	84	28	8.96	9.13	1.06
2	5 min	98	32	10.4	10.8	1.03
3	10 min	99	33	10.5	11.2	1.03
4	15 min	99	33	10.5	11.2	1.03
5	30 min	99	33	10.5	11.2	1.06
6	1 h	99	46	13.0	14.1	1.13
7	2 h	99	83	19.8	21.3	1.20
8	3 h	99	99	22.8	25.6	1.77

^aThe terpolymerizations were performed at a ratio of [TAz]₀/[PA]₀/[BnN(H)Ts]₀/[*t*-BuP₄]₀ = 30/90/1/1 in PO ([TAz]₀ = 1.0 M) at 100°C. ^bDetermined by ¹H NMR in CDCl₃ using integrals of the characteristic signals. ^cCalculated as follows: (M.W. of Initiator) + ([TAz]₀/[I]₀) × conv.(TAz) × (M.W. of TAz + M.W. of PA) + ([PA]₀/[I]₀) × conv.(PA) - ([TAz]₀/[I]₀) × conv.(TAz) × (M.W. of epoxides + M.W. of PA). ^dDetermined by SEC at 35°C in THF (1.0 mL min⁻¹) using PSt standards.

Figure S30. Stacked ^1H NMR spectra (400 MHz, CDCl_3 , 25°C) and SEC traces (THF, 35°C) of the reaction mixture at the ratio of $[\text{Taz}]_0/[\text{PA}]_0/[\text{BnN}(\text{H})\text{Ts}]_0/[t\text{-BuP}_2]_0 = 30/90/1/1$ in PO ($[\text{Taz}]_0 = 1.0 \text{ M}$) at 100°C (entries 22 and 23, Table 1).

Table S15. Terpolymerizations of TAz, PA and PO using $t\text{-BuP}_2^a$

Entry	Time	Conv. (TAz) ^b /%	Conv. (PA) ^b /%	$M_{n,\text{theo}}^c/\text{kg mol}^{-1}$	$M_{n,\text{NMR}}^b/\text{kg mol}^{-1}$	\bar{D}^d
1	2 min	77	25	8.24	8.39	1.07
2	5 min	97	32	10.3	10.7	1.04
3	10 min	99	33	10.5	11.2	1.03
4	15 min	99	33	10.5	11.2	1.06
5	30 min	99	33	10.5	11.2	1.08
6	1 h	99	45	12.8	14.1	1.14
7	2 h	99	67	16.9	17.6	1.16
8	3 h	99	89	20.9	22.1	1.18
9	4 h	99	99	22.8	24.8	1.25

^aThe terpolymerizations were performed at a ratio of $[\text{Taz}]_0/[\text{PA}]_0/[\text{BnN}(\text{H})\text{Ts}]_0/[t\text{-BuP}_2]_0 = 30/90/1/1$ in PO ($[\text{Taz}]_0 = 1.0 \text{ M}$) at 100°C . ^bDetermined by ^1H NMR in CDCl_3 using integrals of the characteristic signals. ^cCalculated as follows: (M.W. of Initiator) + $([\text{Taz}]_0/[\text{I}]_0) \times \text{conv.}(\text{TAz}) \times (\text{M.W. of TAz} + \text{M.W. of PA}) + \{([\text{PA}]_0/[\text{I}]_0) \times \text{conv.}(\text{PA}) - ([\text{Taz}]_0/[\text{I}]_0) \times \text{conv.}(\text{TAz})\} \times (\text{M.W. of epoxides} + \text{M.W. of PA})$. ^dDetermined by SEC at 35°C in THF (1.0 mL min^{-1}) using PSt standards.

Figure S31. Stacked ^1H NMR spectra (400 MHz, CDCl_3 , 25°C) and SEC traces (THF, 35°C) of the reaction mixture at the ratio of $[\text{Taz}]_0/[\text{PA}]_0/[\text{BnN}(\text{H})\text{Ts}]_0/[\text{t-BuP}_2]_0 = 30/90/1/1$ in PO ($[\text{Taz}]_0 = 1.0 \text{ M}$) at 100°C (entries 24 and 25, Table 1).

Figure S32. Stacked ^1H NMR spectra (400 MHz, CDCl_3 , 25°C) and SEC traces (THF, 35°C) of the reaction mixture at the ratio of $[\text{Taz}]_0/[\text{PA}]_0/[\text{BnN}(\text{H})\text{Ts}]_0/[\text{DBU}]_0 = 30/90/1/1$ in PO ($[\text{Taz}]_0 = 1.0 \text{ M}$) at 100°C (entry 26, Table 1).

Figure S33. Stacked ^1H NMR spectra (400 MHz, CDCl_3 , 25°C) and SEC traces (THF, 35°C) of the reaction mixture at the ratio of $[\text{Taz}]_0/[\text{PA}]_0/[\text{BnN}(\text{H})\text{Ts}]_0/[\text{TBD}]_0 = 30/90/1/1$ in PO ($[\text{Taz}]_0 = 1.0 \text{ M}$) at 100°C (entry 27, Table 1).

The block copolymers were analyzed by differential scanning calorimetry (DSC, Figure S57-S58) and thermogravimetric analysis (TGA, Figure S59-S60). Poly($\text{Taz-}alt\text{-PA}$) $_{30}$ - b -poly($\text{PA-}alt\text{-PO}$) $_{60}$ shows a single T_g of 72°C between the T_g of poly($\text{Taz-}alt\text{-PA}$) $_{30}$ (114°C) and poly($\text{PA-}alt\text{-PO}$) $_{60}$ (59°C), indicating a complete mixing of the two alternating blocks, due to the common PA monomer. Decreasing the polymerization degrees of second block increases the T_g value [T_g of Poly($\text{Taz-}alt\text{-PA}$) $_{30}$ - b -poly($\text{PA-}alt\text{-PO}$) $_{30}$ is 78°C], and vice versa [T_g of Poly($\text{Taz-}alt\text{-PA}$) $_{30}$ - b -poly($\text{PA-}alt\text{-PO}$) $_{90}$ is 64°C]. The copolymer catalyzed by other organic bases ($t\text{-BuP}_4$, $t\text{-BuP}_2$, DBU, and TBD) showed lower T_g values ($61\text{-}68^\circ\text{C}$, Figure S60). Compared with homopolymer poly($\text{PA-}alt\text{-PO}$) $_{60}$ ($T_{d\ 5\%} = 327^\circ\text{C}$), all the block copolymers show lower thermal stability ($T_{d\ 5\%} = 251\text{-}266^\circ\text{C}$) (Figure S59-S60).

We added a paragraph (below) and a new Figure S43 in order to better explain the mechanism. Also, we corrected a small mistake in Figure 7. We believe that our work original as well as the catalytic system.

Proposed terpolymerization mechanism

Based on the above experiments, two plausible terpolymerization pathways were proposed (Figure 7 and Figure S43). The two alternating blocks come from the two alternating copolymerization cycles: Taz/PA ROCOP and PA/PO ROCOP, with PA to participate in the two copolymerization cycles. In the first cycle, Taz/PA ROCOP occurs, and kinetic studies show that the PA ring-opening is fast. The resulting phosphazanium carboxylate intermediate ($-\text{COO}/t\text{-BuP}_1\text{-H}^+$, Figure S44) from PA reacts with Taz instead of PO. After ring-opening of Taz, the proton on the $t\text{-BuP}_1\text{-H}^+$ transfers to the new chain-end, forming $-\text{N}(\text{Ts})\text{H}/t\text{-BuP}_1$ (Figure S46), due to the stronger basicity of $-\text{N}(\text{Ts})\text{H}$. The generated amine $-\text{N}(\text{Ts})\text{H}$ rapidly attacks PA instead of PO (Figure 7, cycle 1). The fast proton exchange between the $t\text{-BuP}_1$ and $t\text{-BuP}_1\text{-H}^+$ is conceivable and supported by Figure S46.⁴⁶ Taz/PA ROCOP proceeds quickly until Taz is almost completely consumed. At the end-point of the first copolymerization cycle, slow copolymerization rates were observed at high Taz conversions, and finally, a dormant period. The $t\text{-BuP}_1/t\text{-BuP}_1\text{-H}^+$ complex corresponds to dormant species that can suppress the activation of $t\text{-BuP}_1\text{-H}^+$ to PO (Figure S47). After the complete consumption of Taz, the $t\text{-BuP}_1\text{-H}^+$ released from $t\text{-BuP}_1$ switches to activated species and starts the second copolymerization cycle (PA/PO ROCOP). Similar proton shuttling between the catalysts and growing chain end is considered key to promote selectivity and control for the copolymerization.⁴⁴ For both copolymerization cycles, the phosphazanium shows high monomer selectivity and polymerization controllability.

Figure S43. Plausible mechanistic pathway for *t*-BuP₁-catalyzed terpolymerization of TAz, PA, and PO.

Figure 7. Plausible terpolymerization pathways of chemoselective TAz/PA ROCOP and PA/PO ROCOP by a simple organocatalyst, *t*-BuP₁.

Reviewer #2 (Remarks to the Author):

This manuscript by Hadjichristidis and colleagues describes the selective synthesis of diblock copolymers, where each block is an alternating copolymer. The results are very interesting, as the preparation of diblock dialternating copolymers from a one-pot, one-step method is a significant synthetic challenge. Aziridine/anhydride ring-opening copolymerisation remains relatively underexplored, and I believe the formation of block copolymers from this route will be of significant interest to the community. The selectivity of this route appears to be quite remarkable, as even when performed in neat epoxide, epoxide/anhydride ring-opening copolymerisation does not occur until all of the aziridine is consumed. The addition of further aziridine then switches the reaction back to the aziridine/anhydride copolymerisation, demonstrating the potential to form multiblock copolymers from this route. I believe this article will be of significant interest to the readership of Nature Communications and would recommend this work for publication providing the following points are addressed.

Answer: We thank the Reviewer for the kind comments

The proposed mechanism:

1) After all the aziridine has been consumed, it would be interesting to know why the first epoxide is very slow to open (i.e. why there is a “dormant period” between the aziridine/anhydride and epoxide/anhydride polymerisations). This delay a little surprising – it seems that the first equivalent of epoxide is slow to ring-open (from an opened anhydride chain end), yet subsequent epoxide insertions (into the opened anhydride chain ends) are much faster. Can the authors provide any insight into this observation?

Answer: The mechanism was better explained and shown in Figure 7 and S43. NMR titration experiments were performed to verify the mechanism (Figure S44-S56). The corresponding description has been included in the manuscript.

All the NMR titration experiments were re-done in Young’s NMR tubes using anhydrous toluene-*d*₈ solvent under argon (avoid interference of CHCl₃ on *t*-BuP₁).

The corresponding mechanism in the manuscript is described as followed:

Proposed terpolymerization mechanism

Based on the above experiments, two plausible terpolymerization pathways are proposed (Figure 7 and Figure S43). The two alternating blocks come from the two alternating copolymerization cycles: TAz/PA ROCOP and PA/PO ROCOP, with PA to participate in both copolymerization cycles. In the first cycle, TAz/PA ROCOP takes place, and kinetic studies show that the PA ring-opening is fast. The resulting phosphazanium carboxylate intermediate (-COO⁻/*t*-BuP₁-H⁺, Figure S44) from PA reacts with TAz instead of PO. After ring-opening of TAz, the proton on the *t*-BuP₁-H⁺ is transferred to the new chain-end, forming -N(Ts)H/*t*-BuP₁ (Figure S46), due to the stronger basicity of -N(Ts)⁻. The generated amine -N(Ts)H rapidly attacks PA

instead of PO (Figure 7, cycle 1). The fast proton exchange between the $t\text{-BuP}_1$ and $t\text{-BuP}_1\text{-H}^+$ is plausible and supported by Figure S46.⁴⁷ TAz/PA ROCOP proceeds quickly until TAz is almost completely consumed. At the end-point of the first copolymerization cycle, slow copolymerization rates were observed at high TAz conversions, and finally, a dormant period. The $t\text{-BuP}_1/t\text{-BuP}_1\text{-H}^+$ complex corresponds to dormant species that can suppress the activation of $t\text{-BuP}_1\text{-H}^+$ to PO (Figure S47). After the complete consumption of TAz, the $t\text{-BuP}_1\text{-H}^+$ released from $t\text{-BuP}_1$ switches to activated species and starts the second copolymerization cycle (PA/PO ROCOP). Similar proton shuttling between the catalysts and growing chain end is considered key to promote selectivity and control for the copolymerization.⁴⁵ For both copolymerization cycles, the phosphazene shows high monomer selectivity and polymerization controllability.

Figure S43. Plausible mechanistic pathway for $t\text{-BuP}_1$ -catalyzed terpolymerization of TAz, PA, and PO.

Figure 7. Plausible terpolymerization pathways of chemoselective TAz/PA ROCOP and PA/PO ROCOP by a simple organocatalyst, $t\text{-BuP}_1$.

The explanation for the dormant period is given below:

The first copolymerization of aziridine/anhydride is a first-order dependence with the aziridine concentration and a zero-order dependence with the anhydride concentration. When the aziridine is almost consumed, the low aziridine concentration results in a slow rate of aziridine/anhydride copolymerization. The chain-end is a mixture of $\text{-COO}^-/t\text{-BuP}_1\text{H}^+$ and $(\text{Ts})\text{NH}/t\text{-BuP}_1$. Figure S44 (new) supports that $t\text{-BuP}_1$ deprotonates benzoic acid. Figure S46 (new) supports that N^- deprotonates $t\text{-BuP}_1\text{H}^+$. The combination of $t\text{-BuP}_1\text{H}^+$ and $t\text{-BuP}_1$ cannot activate epoxide, which is the dormant period. Figure S47 (new) supports that $t\text{-BuP}_1\text{H}^+$ can activate epoxide by hydrogen bonding. After the full consumption of aziridines, the chain-end $\text{COO}^-/t\text{-BuP}_1\text{H}^+$ without any $t\text{-BuP}_1$ can activate epoxide, which starts the second copolymerization of anhydride/epoxide. The dormant period is the process of complete consumption of trace aziridine (although the full conversion of aziridine was detected by ^1H NMR, there is trace aziridine in the reaction to produce $(\text{Ts})\text{NH}/t\text{-BuP}_1$, suppressing the activation of $t\text{-BuP}_1\text{H}^+$ to epoxide).

Besides, for both blocks, the molecular weights match with the theoretical value, and the molecular weight distributions remain narrow, indicating a living character for the two alternating copolymerizations. The living character ensures a fast initiation of the first epoxide instead of a slow initiation, indicating that the dormant period is the progress of a slow rate of aziridine/anhydride copolymerization.

Figure S44. Stacked ^1H NMR (600 MHz, $\text{toluene-}d_8$, 25°C) for (a) $t\text{-BuP}_1$, (b) BA, and (c) $t\text{-BuP}_1 / \text{BA} = 1/1$.

Figure S46. Stacked ¹H NMR (600 MHz, toluene-*d*₈, 25°C) for (a) *t*-BuP₁, (b) BA, (c) *t*-BuP₁ / BA = 1/1, and (d) *t*-BuP₁ / BA / TAZ = 1/1/1. (d: add *t*-BuP₁, BA, and TAZ into the NMR tube)

Figure S47. Stacked ¹H NMR (600 MHz, toluene-*d*₈, 25°C) for (a) *t*-BuP₁, (b) BA, (c) *t*-BuP₁ / BA = 1/1, (d) *t*-BuP₁ / PO = 1/1, and (e) *t*-BuP₁ / BA / PO = 1/1/1.

2) Figure S36 is very interesting. Could the significant shift observed for the organocatalyst (*t*BuP₁) in the absence/presence of the aziridine be because the organocatalyst preferentially interacts with the aziridine instead of the epoxide as shown (top), which may activate the aziridine towards ring-opening? Is there any possibility that proton transfer from the initiator to the aziridine could occur (and if so, enhance ring-opening), instead of proton transfer to the organocatalyst? I would add a ¹H NMR spectra of the combination of initiator and aziridine, to see whether any proton transfer could be observed. It may also be helpful to include the ¹H NMR of the catalyst and initiator as separate species, to confirm the proposed proton transfer between these two species. Note that in this figure, there are currently two spectra labelled as b.

Answer: We performed the experiments the Reviewer suggested.

We mixed equivalent *t*-BuP₁ and aziridine in toluene-*d*₈ in Young's NMR tube under argon, as shown in Figure S45. There is almost no shift observed for the *t*-BuP₁ in the absence/presence of aziridine (Figure S45, new). The results indicate that the *t*-BuP₁ cannot abstract the proton from aziridine.

-We mixed equivalent *t*-BuP₁ and benzoic acid in toluene-*d*₈ in Young's NMR tube under argon, as shown in Figure S44, new (in answer 1). The proton transfers to the high field from benzoic acid COOH (12.51 ppm) to *t*-BuP₁H⁺ (10.76 ppm). The protons of benzene on benzoic acid transfer to low field (8.08 to 8.64 ppm; 7.03 to 7.34 ppm). The protons on *t*-BuP₁ transfer to the high field (2.52 to 2.38 ppm; 1.45 to 1.33 ppm). All the shifts indicate that the *t*-BuP₁ can deprotonate the proton on benzoic acid. The basicity of *t*-BuP₁ is higher than the basicity of COO⁻. The *t*-BuP₁ cannot abstract the proton on aziridine. Hence, COO⁻ cannot abstract the proton on aziridine.

-We mixed equivalent *t*-BuP₁, benzoic acid, and aziridine in toluene-*d*₈ in Young's NMR tube under argon, as shown in Figure S46, new (in answer 1). The proton transfers to the high field from benzoic acid COOH (12.51 ppm) to *t*-BuP₁H⁺ (10.76 ppm), and to (Ts)NH (9.78 ppm). The constant shift to the high field indicates the deprotonation from COOH to *t*-BuP₁H⁺ to (Ts)NH. The protons on *t*-BuP₁ transfer to the high field (2.52 to 2.38 ppm; 1.45 to 1.325 ppm), and then to the low field (2.41 ppm; 1.332 ppm), indicating the hydrogen bonding between the *t*-BuP₁ and *t*-BuP₁H⁺.

Figure S45. Stacked ¹H NMR (600 MHz, toluene-*d*₈, 25°C) for (a) *t*-BuP₁, (b) TAz, (c) *t*-BuP₁ / TAz = 1/1, and (d) *t*-BuP₁ / TAz = 1/1 after 24 h.

3) The proposed catalytic cycle shown in Figure 7 needs some attention. The aziridine and epoxide (top left and top right) should be shown as entering into the catalytic cycles, rather than as intermediates formed from the active opened anhydride/organocatalyst species (centre). The charges do not balance as shown (e.g. should the organocatalyst have a proton on the left hand side structure with the ring-opened aziridine)? The bottom figures should also include the organocatalyst to balance the equations. I would also recommend removing "*" from the bottom figures (or explaining what this symbol refers to), as "*" is sometimes used to denote an active chain end in the polymer literature.

Answer: We changed Figure 7 according to Reviewer's suggestion. We also removed "*" in MS and SI.

Figure 7. Plausible terpolymerization pathways of chemoselective TAz/PA ROCOP and PA/PO ROCOP by a simple organocatalyst, *t*-BuP₁.

Additional points:

4) “The benzene signals of TAz” - change benzene for aromatic.

Answer: Corrected benzene to aromatic.

5) I would recommend rephrasing “The molecular weight of the copolymer increases with the reaction time, in agreement with the theoretical molecular weight of poly(TAz-*alt*-PA) (Figure 6b).” This statement may be misleading; the copolymer MW cannot be directly compared to the theoretical weight, as the SEC values were determined by calibration against polystyrene standards.

Answer: The molecular weights in Figure 6b were calculated by ¹H NMR rather than SEC. To avoid misunderstanding, we corrected to “The molecular weight of the copolymer calculated by ¹H NMR increases with the reaction time, in agreement with the theoretical molecular weight of poly(TAz-*alt*-PA) (Figure 6b).”

6) “The decreased temperature slows down the reaction rates of both the first-stage (k_{obs}, first-stage = 11.2547 M⁻¹h⁻¹ in 100o C; k_{obs}, first-stage = 0.4734 M⁻¹h⁻¹ in 60o C) and the

second-stage copolymerization ($k_{obs, \text{ second-stage}} = 0.4500 \text{ M}^{-1}\text{h}^{-1}$ in 100o C; $k_{obs, \text{ second-stage}} = 0.0682 \text{ M}^{-1}\text{h}^{-1}$ in 60o C) (Figure S31-S34).” It may be helpful to show how these numbers were derived from Figures S31-34. Should one of these figures also show the second stage copolymerisation at 60oC?

Answer: The numbers were derived from the following equations, given also in SI:

$$-\frac{d[PA]}{dt} = k_{obs}$$

$$-\int d[PA] = k_{obs} \int dt$$

$$[PA]_0 - [PA]_t = k_{obs} \times t$$

$$\frac{[PA]_0 - [PA]_t}{t} = k_{obs}$$

$$\frac{\text{conv. \%} \div 100}{t \div [PA]_0} = k_{obs}$$

These numbers ($k_{obs, \text{ first-stage}} = 11.7196 \text{ mol L}^{-1}\text{h}^{-1}$ in 100°C; $k_{obs, \text{ first-stage}} = 0.4773 \text{ mol L}^{-1}\text{h}^{-1}$ in 60°C; $k_{obs, \text{ second-stage}} = 0.4500 \text{ mol L}^{-1}\text{h}^{-1}$ in 100°C; $k_{obs, \text{ second-stage}} = 0.0682 \text{ mol L}^{-1}\text{h}^{-1}$ in 60°C) can be calculated and shown in the legend (Figure S38-S41).

Figure S41 is the second stage of copolymerization at 60°C.

7) “All the reactivity ratios support the following: copolymerization rate of TAz/PA > homopolymerization rate of TAz > copolymerization rate of TAz/PO > homopolymerization rate of PA/PO.” – should homopolymerization rate of PA/PO be copolymerisation instead?

Answer: “Homopolymerization rate of PA/PO” means “homopolymerization rate of PA or homopolymerization rate of PO”. To avoid misunderstanding, we have changed to “homopolymerization rate of PA or homopolymerization rate of PO”.

8) “Furthermore, with the use of multifunctional initiators, as in the case of the synthesized pentablock, this methodology opens new horizons to dialternating-based complex macromolecular architectures with unprecedented properties.” It would be helpful to include evidence to support the pentablock structure (or to make it clear that this assignment is somewhat speculative), as it is currently unclear whether initiation occurs from one or both ends of the diols.

Answer: Indeed, this methodology opens new horizons to CMA since all hydroxyl groups can initiate the copolymerization. The prove is that we have not observed any methylene signal peak of the 1,4-benzenedimethanol (4.73 ppm, Figure S19) on the copolymers (Figure S18), indicating that both hydroxyl groups on the 1,4-benzenedimethanol can initiate the copolymerization.

In order to clarify this point we added the following paragraph in the MS:

We did not observe any methylene signal peak of the 1,4-benzenedimethanol (4.73 ppm, Figure S19) on the copolymers (Figure S18), indicating that both hydroxyl groups on the 1,4-benzenedimethanol can initiate the copolymerization.

ESI

9) The experimental states “1,4-Dioxane, propylene oxide (PO), and butylene oxide (BO) were dried over calcium hydride, over n-butyllithium overnight, distilled, and stored in a glovebox before use.” Does n-butyllithium not react with (ring-open) the epoxides (this has been documented in literature)?

Answer: This is correct, n-butyllithium can react with epoxides but very slowly. In order to clarify this point we corrected our sentence. As follows: 1,4-Dioxane, propylene oxide (PO), and butylene oxide (BO) were dried over calcium hydride overnight, distilled, and then dried over n-butyllithium for 2 h, distilled, and then stored in a glovebox before use.

10) The reactions were typically performed at boiling points much higher than the boiling points of the epoxides. It would be important to acknowledge the pressure build-up (and associated safety aspects) of performing the reactions under these conditions. It would also be important to explain how the reaction was set up when using ethylene oxide, which is a gas at room temperature.

Answer: We can estimate the internal pressure according to the ideal gas law ($PV = nRT$; P is pressure, V is volume, n is amount of substance, R is ideal gas constant, and T is temperature). The thick Schlenk reactor can withstand 6 atm.

We add the following paragraph to clarify this point. “Entry 11 procedure (Table 1) is following as an example. In an argon-filled glovebox, PA (0.264 g, 1.8 mmol), TAz (0.116 g, 0.6 mmol), and initiator (N-benzyl-4-methylbenzenesulfonamide) (0.0052 g, 0.02 mmol) were charged into a 100 mL thick Schlenk reactor (ideal gas law is used to estimate internal pressure). The reactor was then put in the refrigerator (-20°C) for 60 min, and then anhydrous EO (0.6 mL) was introduced into the reactor. This step was performed in the refrigerator (-20°C) in the glovebox. After 5 min of stirring, the reactor was put in the refrigerator (-20°C) for 60 min. *t*-BuP₁ (4.6 μL, 0.02 mmol)

was added into the mixture. After sealing, the reactor was taken out of the glovebox and stirred at 100°C to start the polymerization. The polymerization was monitored by ¹H NMR and SEC. The produced polymer was precipitated by the slow addition of the diluted reaction solution (CHCl₃) to an excess of methanol. The precipitation process was repeated three times, and then the terpolymer was dried in a vacuum oven at 40°C overnight.”

11) Table S10 footnote e, it would be helpful to clarify when TAz was added.

Answer: Corrected to “The addition of extra TAz” (Table S18).

12) Fig S31 – It would help to add a few extra data points if possible.

Answer: Two more points were added, as shown in Figure S38.

Figure S38. Zero-order plot of PA conversion versus time at the first stage copolymerization at 100°C ([PA]₀ = 3.0 M) ($k_{\text{obs}} = 6.5109/100 \cdot 60 \cdot 3 = 11.7196 \text{ mol L}^{-1}\text{h}^{-1}$).

Reviewer #3 (Remarks to the Author):

The results in this manuscript represent a continuous work on the ring-opening copolymerization (ROCOP) of aziridines and cyclic anhydrides (Angew. Chem. Int. Ed. 2021, 60, 6949–6954), in which the authors extended the process to a “one pot/one step” terpolymerization employing aziridines, cyclic anhydrides and epoxides. Interestingly, the authors found that the copolymerization of aziridines and cyclic anhydrides is thermodynamically favored with respect to epoxide/anhydride copolymerization, leading to di- or tri-blocks copolymers (when using a bifunctional initiator) with perfectly alternating segments of aziridine/anhydride and epoxide/anhydride, thus affording poly(ester amides)-b-poly(ester) block copolymers. Impressively, the latter are formed without any incorporation of epoxide into the aziridine/anhydride segments despite performing the terpolymerization in neat epoxide. The authors provide compelling evidences for perfect selectivity in monomer enchainment, using NMR, IR, MALDI-ToF and SEC techniques, and supported by DFT calculations of thermodynamic parameters. Furthermore, the authors show that by feeding an additional portion of aziridine to the reaction mixture during the second ROCOP reaction (epoxide/anhydride), the latter is inhibited and a shift to the first ROCOP cycle occurs until complete consumption of the aziridine monomer. Such process has therefore the potential to lead to tailor-made block copolymers of alternating poly(ester amides) and poly(ester) segments. The NMR results are comprehensive and validate the sequence of ROCOP reactions, including a dormant stage between the two phases. The IR and SEC data complement the NMR results, further corroborating the selectivity in monomer incorporation and the living nature of the catalytic system. Nevertheless, the authors do not provide any physical properties of the polymers obtained (with the exception of a single Tg value) which makes it hard to evaluate whether these polymers have valuable industrially-relevant potential, as stated in the introduction and conclusions. In addition, there are a few issues that the authors should address:

Answer: We thank the Reviewer for his kind words.

1) In the context of switchable “one pot/one step” polymerization the work by Williams and coworkers should be cited (Catal. Sci. Technol., 2021,11, 1737-1745; ; Chem. Commun., 2019, 55, 7315).

Answer: These works were added as ref 33 and ref 34.

2) Page 4: the authors should refer to the initial monomers feed ratio when discussing the % conversions in order to avoid confusion. In this context, the authors did not explain why a 1:3 Taz/PA ratio was chosen (instead of 1:2 for example).

Answer: The initial monomers feed ratio was added in the manuscript when discussing the % conversions: TAz and PA conversions reached 91% and 30% within 45 min ($[TAz]_0/[PA]_0 = 1/3$), respectively.

Other initial monomers feed ratio (TAz/PA = 1:2 and 1:4) was added in the manuscript (entries 28-31 Table 1) and Supporting Information (Figure S34 and S35, Table S16 and S17). The dialternating diblock copolymers can be successfully synthesized under different monomer ratios (TAz/PA = 1:2, 1:3, and 1:4). The corresponding description was shown in the manuscript, as followed: Initial monomers feed ratios (TAz/PA) were changed to 1/2 and 1/4 in the terpolymerization (entries 28-31 in Table 1 and Figure S34-S35). The 1H NMR and SEC traces (Figure S34-S35) confirm the successful synthesis of diblock dialternating terpolymers.

Table S16. Terpolymerizations of TAz, PA and PO using t -BuP₁^a

Entry	Time	Conv. (TAz) ^b /%	Conv. (PA) ^b /%	$M_{n,theo}$ ^c /kg mol ⁻¹	$M_{n,NMR}$ ^b /kg mol ⁻¹	\bar{D} ^d
1	2 min	25	12	2.85	2.96	1.07
2	5 min	63	21	6.79	6.98	1.04
3	10 min	94	47	10.0	10.8	1.03
4	15 min	99	49	10.5	11.2	1.03
5	30 min	99	49	10.5	11.2	1.03
6	1 h	99	52	10.9	11.9	1.06
7	2 h	99	69	13.0	13.7	1.04
8	3 h	99	88	15.3	15.9	1.04
9	4 h	99	99	16.7	18.2	1.04

^aThe terpolymerizations were performed at a ratio of $[TAz]_0/[PA]_0/[BnN(H)Ts]_0/[t-BuP_1]_0 = 30/60/1/1$ in PO ($[TAz]_0 = 1.0$ M) at 100°C. ^bDetermined by 1H NMR in CDCl₃ using integrals of the characteristic signals. ^cCalculated as follows: (M.W. of Initiator) + ($[TAz]_0/[I]_0$) × conv.(TAz) × (M.W. of TAz + M.W. of PA) + ($[PA]_0/[I]_0$) × conv.(PA) - ($[TAz]_0/[I]_0$) × conv.(TAz) × (M.W. of epoxides + M.W. of PA). ^dDetermined by SEC at 35°C in THF (1.0 mL min⁻¹) using PSt standards.

Figure S34. Stacked ^1H NMR spectra (400 MHz, CDCl_3 , 25°C) and SEC traces (THF, 35°C) of the reaction mixture at the ratio of $[\text{Taz}]_0/[\text{PA}]_0/[\text{BnN}(\text{H})\text{Ts}]_0/[\text{t-BuP}_1]_0 = 30/60/1/1$ in PO ($[\text{Taz}]_0 = 1.0 \text{ M}$) at 100°C (entries 28 and 29, Table 1).

Table S17. Terpolymerizations of TAz, PA and PO using t-BuP_1 ^a

Entry	Time	Conv. (TAz) ^b /%	Conv. (PA) ^b /%	$M_{n,\text{theo}}^c/\text{kg mol}^{-1}$	$M_{n,\text{NMR}}^b/\text{kg mol}^{-1}$	\bar{D}^d
1	2 min	36	9	4.00	4.31	1.07
2	5 min	77	19	8.24	8.98	1.05
3	10 min	99	24	10.5	11.2	1.03
4	15 min	99	24	10.5	11.2	1.03
5	30 min	99	24	10.5	11.2	1.03
6	1 h	99	30	11.8	13.1	1.06
7	2 h	99	35	13.1	14.2	1.04
8	3 h	99	47	16.0	16.9	1.04
9	4 h	99	58	18.8	20.1	1.04
10	6 h	99	88	26.2	27.8	1.03
11	8 h	99	99	28.9	30.0	1.02

^aThe terpolymerizations were performed at a ratio of $[\text{Taz}]_0/[\text{PA}]_0/[\text{BnN}(\text{H})\text{Ts}]_0/[\text{t-BuP}_1]_0 = 30/120/1/1$ in PO ($[\text{Taz}]_0 = 1.0 \text{ M}$) at 100°C . ^bDetermined by ^1H NMR in CDCl_3 using integrals of the characteristic signals. ^cCalculated as follows: $(\text{M.W. of Initiator}) + ([\text{Taz}]_0/[\text{I}]_0) \times \text{conv.}(\text{TAz}) \times (\text{M.W. of TAz} + \text{M.W. of PA}) + \{([\text{PA}]_0/[\text{I}]_0) \times \text{conv.}(\text{PA}) - ([\text{Taz}]_0/[\text{I}]_0) \times \text{conv.}(\text{TAz})\} \times (\text{M.W. of epoxides} + \text{M.W. of PA})$. ^dDetermined by SEC at 35°C in THF (1.0 mL min^{-1}) using PSt standards.

Figure S35. Stacked ^1H NMR spectra (400 MHz, CDCl_3 , 25°C) and SEC traces (THF, 35°C) of the reaction mixture at the ratio of $[\text{TAz}]_0/[\text{PA}]_0/[\text{BnN}(\text{H})\text{Ts}]_0/[\text{t-BuP}_1]_0 = 30/120/1/1$ in PO ($[\text{TAz}]_0 = 1.0 \text{ M}$) at 100°C (entries 30 and 31, Table 1).

3) Page 9: the NMR shifts ascribed to the poly(PA-alt-PO) segment are methine and methylene signals, and not “methylene and methyl group” as noted for j and k in Figure 4a.

Answer: Corrected to “methine and methylene signals”.

4) Page 9: “There are no ether linkage signals” instead of “There is no ether linkages signal”, as these would be two signals.

Answer: Corrected to “There are no ether linkage signals”.

5) Page 9: the authors state that “The characteristic signal peak of the carbonyl group from poly(PA-alt-PO) was observed at 167.0 ppm”, however there are two carbonyl signals in the ^{13}C NMR spectrum ascribed to both carbonyl groups of PA which are de-symmetrized upon copolymerization with PO.

Answer: Corrected to “The characteristic signal peaks of the carbonyl group from poly(PA-*alt*-PO) were observed at 167.0 and 168.8 ppm”

6) Page 10: Figure 4a is too small which makes extremely hard to follow the signal annotations.

Answer: We made it bigger.

Figure 4. (a) ¹H NMR and (b) DOSY NMR spectra of poly(TAz-*alt*-PA)-*b*-poly(PA-*alt*-PO).

7) Page 13: the catalytic cycles are drawn wrong; the monomers should not be part of the cycle but rather enter the cycle by external arrows. Further, the amine anion (and corresponding alkoxide in the cycle 2) should react with PA (monomer B) prior to forming a polymeryl chain with a carboxylate end group, in contrast to the drawing in which it looks as if a polymer (without any reactive end group) is reacting with the monomer. Finally, the same overall charge should be maintained throughout the cycle (the catalytic base should balance the reactive anionic end groups as in the conjugation of the two cycles).

Answer: The new corrected Figure 7 was shown in the manuscript.

Figure 7. Plausible terpolymerization pathways of chemoselective TAz/PA ROCOP and PA/PO ROCOP by a simple organocatalyst, *t*-BuP₁.

8) Page 16: the authors state that the steric hindrance of the bulkier epoxides is responsible for the incomplete terpolymerization and a decrease in the alternating nature of Taz/PA. I would expect a bulkier epoxide to increase the selectivity in Taz/PA insertion given the slower rate of ring-opening observed by the authors. In addition, based on the NMR figure it seems that PA was completely converted so the polymer mixture obtained might be the consequence of inefficient initiation of the second ROCOP process. The authors did not provide any SEC data (M_n vs. theoretical M_n , PDI) that will allow to hypothesize about this issue.

Answer: Theoretical M_n , $M_{n,SEC}$ and \bar{D} were added in SI (Table S10-S13). Due to poor alternating nature, it is difficult to calculate TAz and PA conversion, but only complete conversion, as well as M_n by NMR the reason we compare with $M_{n,SEC}$. Nevertheless, the $M_{n,SEC}$ is obviously less than the theoretical M_n , and the retention volume is less than the previous copolymer (in PO), indicating a decrease in the alternating nature of TAz/PA. These epoxides maybe not the suitable solvent for alternating copolymerization of TAz/PA. In the second copolymerization, incomplete block copolymers were observed, indicating an inefficient initiation of the second cycle. The steric hindrance may be the cause of the inefficient initiation.

Table S10. Copolymerizations of TAz, PA and SO using *t*-BuP₁^a

Entry	Time	Conv. (TAz) ^b /%	Conv. (PA) ^b /%	$M_{n,theo}^c$ /kg mol ⁻¹	$M_{n,SEC}^d$ /kg mol ⁻¹	\bar{D}^d
1	5 min				4.02	1.14
2	10 min				6.19	1.10
3	15 min				7.27	1.08
4	30 min	99		10.5	7.53	1.08
5	1 h	99			7.60	1.09
6	2 h	99			8.57	1.15
7	3 h	99	99	26.6	9.97	1.03; 1.08

^aThe terpolymerizations were performed at a ratio of [TAz]₀/[PA]₀/[Bn(Ts)NH]₀/[*t*-BuP₁]₀ = 30/90/1/1 in SO ([TAz]₀ = 1.0 M) at 100°C. ^bDetermined by ¹H NMR in CDCl₃. ^cCalculated as follows: (M.W. of Initiator) + ([TAz]₀/[I]₀) × conv. (TAz) × (M.W. of TAz + M.W. of PA) + {([PA]₀/[I]₀) × conv. (PA) - ([TAz]₀/[I]₀) × conv. (TAz)} × (M.W. of epoxides + M.W. of PA). ^dDetermined by SEC at 35°C in THF (1.0 mL min⁻¹) using PSt standards.

Figure S25. Stacked ¹H NMR spectra (400 MHz, CDCl₃, 25°C) and SEC traces (THF, 35°C) of the reaction mixture at a ratio of [TAz]₀/[PA]₀/[Bn(H)Ts]₀/[*t*-BuP₁]₀ = 30/90/1/1 in SO ([TAz]₀ = 1.0 M) at 100°C.

Table S11. Copolymerizations of TAz, PA and EPP using *t*-BuP₁^a

Entry	Time	Conv. (TAz) ^b /%	Conv. (PA) ^b /%	$M_{n,theo}^c$ /kg mol ⁻¹	$M_{n,SEC}^d$ /kg mol ⁻¹	\bar{D}^d
1	5 min				6.37	1.06
2	10 min				6.68	1.06
3	15 min				7.08	1.06
4	30 min	99		10.5	8.18	1.12

5	1 h	99			16.7; 7.00	1.03; 1.06
6	2 h	99	99	28.4	19.9; 7.76	1.06; 1.05

^aThe terpolymerizations were performed at a ratio of $[Taz]_0/[PA]_0/[Bn(Ts)NH]_0/[t-BuP_1]_0 = 30/90/1/1$ in EPP ($[Taz]_0 = 1.0$ M) at 100°C . ^bDetermined by ^1H NMR in CDCl_3 . ^cCalculated as follows: $(\text{M.W. of Initiator}) + ([Taz]_0/[I]_0) \times \text{conv.}(Taz) \times (\text{M.W. of Taz} + \text{M.W. of PA}) + \{([PA]_0/[I]_0) \times \text{conv.}(PA) - ([Taz]_0/[I]_0) \times \text{conv.}(Taz)\} \times (\text{M.W. of epoxides} + \text{M.W. of PA})$. ^dDetermined by SEC at 35°C in THF (1.0 mL min^{-1}) using PSt standards.

Figure S26. Stacked ^1H NMR spectra (400 MHz, CDCl_3 , 25°C) and SEC traces (THF, 35°C) of the reaction mixture at a ratio of $[Taz]_0/[PA]_0/[BnN(H)Ts]_0/[t-BuP_1]_0 = 30/90/1/1$ in EPP ($[Taz]_0 = 1.0$ M) at 100°C .

Table S12. Copolymerizations of Taz, PA and NBGE using $t\text{-BuP}_1$ ^a

Entry	Time	Conv. (Taz) ^b /%	Conv. (PA) ^b /%	$M_{n,theo}$ ^c /kg mol^{-1}	$M_{n,SEC}$ ^d /kg mol^{-1}	\bar{D} ^d
1	5 min				5.15	1.08
2	10 min				5.52	1.08
3	15 min				5.65	1.08
4	30 min				5.89	1.08
5	1 h	99		10.5	7.00	1.14
6	2 h	99			7.44	1.08; 1.06
7	3 h	99			16.3; 6.28	1.11; 1.08
8	4 h	99	99	27.2	19.5; 6.34	1.16; 1.07

^aThe terpolymerizations were performed at a ratio of $[TAz]_0/[PA]_0/[Bn(Ts)NH]_0/[t-BuP_1]_0 = 30/90/1/1$ in NBGE ($[TAz]_0 = 1.0$ M) at 100°C. ^bDetermined by ¹H NMR in CDCl₃. ^cCalculated as follows: (M.W. of Initiator) + ($[TAz]_0/[I]_0$) × conv.(TAz) × (M.W. of TAz + M.W. of PA) + ($[PA]_0/[I]_0$) × conv.(PA) - ($[TAz]_0/[I]_0$) × conv.(TAz) × (M.W. of epoxides + M.W. of PA). ^dDetermined by SEC at 35°C in THF (1.0 mL min⁻¹) using PSt standards.

Figure S27. Stacked ¹H NMR spectra (400 MHz, CDCl₃, 25°C) and SEC traces (THF, 35°C) of the reaction mixture at a ratio of $[TAz]_0/[PA]_0/[BnN(H)Ts]_0/[t-BuP_1]_0 = 30/90/1/1$ in NBGE ($[TAz]_0 = 1.0$ M) at 100°C.

Table S13. Copolymerizations of TAz, PA and CHO using *t*-BuP₁^a

Entry	Time	Conv. (TAz) ^b /%	Conv. (PA) ^b /%	$M_{n,theo}^c$ /kg mol ⁻¹	$M_{n,SEC}^d$ /kg mol ⁻¹	\bar{D}^d
1	5 min				2.12	1.19
2	10 min				2.84	1.14
3	15 min				3.07	1.14
4	30 min	99		10.5	3.21	1.14
5	1 h	99			3.08	1.15
6	2 h	99			3.15	1.17
7	3 h	99			3.29	1.25
8	5 h	99	99	25.3	14.5; 3.36	1.10; 1.15

^aThe terpolymerizations were performed at a ratio of $[TAz]_0/[PA]_0/[Bn(Ts)NH]_0/[t-BuP_1]_0 = 30/90/1/1$ in CHO ($[TAz]_0 = 1.0$ M) at 100°C. ^bDetermined by ¹H NMR in CDCl₃. ^cCalculated as follows: (M.W. of Initiator) + ($[TAz]_0/[I]_0$) × conv.(TAz) × (M.W. of TAz + M.W. of PA) + ($[PA]_0/[I]_0$) × conv.(PA) - ($[TAz]_0/[I]_0$) × conv.(TAz) × (M.W. of epoxides + M.W. of PA). ^dDetermined by SEC at 35°C in THF (1.0 mL min⁻¹) using PSt standards.

Figure S28. Stacked ^1H NMR spectra (400 MHz, CDCl_3 , 25°C) and SEC traces (THF, 35°C) of the reaction mixture at a ratio of $[\text{Taz}]_0/[\text{PA}]_0/[\text{BnN(H)Ts}]_0/[\text{t-BuP}_1]_0 = 30/90/1/1$ in CHO ($[\text{Taz}]_0 = 1.0 \text{ M}$) at 100°C .

9) SI, page 2: Overall, the experimental data is well presented, however the authors neglected to explain how exactly they acquired the conversion/kinetic data. The reactions were performed at a high temperature (100°C), which exceeds the boiling point of THF and most employed epoxides. Thus, the authors used a Schlenk reactor, which I presume could support the internal pressure formed during the reaction, and performed the reaction outside the glove box. Therefore, the authors should comment on how the data for the polymerization was acquired. Did they withdraw aliquots from a single reaction or did they performed several processes simultaneously, quenching the each of the reactions at different reaction times? If the latter was performed, did the reactions were performed in triplicates? Are the presented data points average values (in which case an error should be reported as well)?

Answer: We conducted a series of parallel experiments that monitored by ^1H NMR spectra by quenching aliquots for convincing experimental results. These reactions were repeated three-time, and all the results were very close. We added relevant descriptions in the manuscript, as followed:

Since the low-boiling point of PO (34°C) easily causes changes in the reaction concentration, we conducted a series of parallel experiments monitored by ^1H NMR spectra by quenching aliquots for convincing experimental results (repeat all results three times).

10) SI: Did the authors use an internal standard for the kinetic measurements in order to determine the accuracy of the concentrations and resulting conversions? If they did, it should be mentioned in the SI section.

Answer: We did not use an internal standard for the kinetic measurements. The conversions were calculated by ^1H NMR integral of characteristic signals of monomers and polymers.

11) SI: The in-situ FTIR instrument does not appear in the instrumentation section of the SI and the authors did not comment on the experimental details of its usage. Did the authors use an instrument equipped with an internal probe that could be inserted into the reaction mixture? The authors should elaborate on the experimental setup and data acquisition methods.

Answer: We added the *in-situ* FTIR instrument description in the manuscript. "*In-situ* Fourier-transform infrared spectroscopy (FTIR) study was conducted using a ReactIR 45m (Mettler Toledo). The samples were collected every 1 min, and each spectrum was scanned 256 times."

Yes, the *in-situ* FTIR instrument is equipped with an internal probe that could be inserted into the reaction mixture.

12) SI, page 25: condition "b" appears twice in Figure S36. In addition, this NMR evidence for the inhibition of the propagating carboxylate by residual TAz is not compelling enough. Upon the addition of TAz the chemical shift of the carboxylate's alpha proton is only slightly affected, thus rendering its supposed shielding doubtful. In addition, similar interactions could be envisioned in the copolymerization of EO a between the carboxylate and the methylene protons of the epoxide (which is in a large excess), which would lead to epoxide ring-opening during a polymerization process. However, a similar dormant stage appears in the ROCOP of EO as well. The authors should consider to revisit this experiment, possibly performing several control experiments in which the carboxylate will be treated separately with TAz and EO, followed by titration of each reaction with the opposed monomer.

Answer: The mechanism was explained more and shown in Figure 7 and Figure S43.

The corresponding mechanism in the manuscript is described as follows:

Proposed terpolymerization mechanism

Based on the above experiments, two plausible terpolymerization pathways are proposed (Figure 7 and Figure S43). The two alternating blocks come from the two alternating copolymerization cycles: TAz/PA ROCOP and PA/PO ROCOP, with PA to participate in both

copolymerization cycles. In the first cycle, TAz/PA ROCOP takes place, and kinetic studies show that the PA ring-opening is fast. The resulting phosphazanium carboxylate intermediate ($-\text{COO}^-/t\text{-BuP}_1\text{-H}^+$, Figure S44) from PA reacts with TAz instead of PO. After ring-opening of TAz, the proton on the $t\text{-BuP}_1\text{-H}^+$ is transferred to the new chain-end, forming $-\text{N}(\text{Ts})\text{H}/t\text{-BuP}_1$ (Figure S46), due to the stronger basicity of $-\text{N}(\text{Ts})^-$. The generated amine $-\text{N}(\text{Ts})\text{H}$ rapidly attacks PA instead of PO (Figure 7, cycle 1). The fast proton exchange between the $t\text{-BuP}_1$ and $t\text{-BuP}_1\text{-H}^+$ is plausible and supported by Figure S46.⁴⁷ TAz/PA ROCOP proceeds quickly until TAz is almost completely consumed. At the end-point of the first copolymerization cycle, slow copolymerization rates were observed at high TAz conversions, and finally, a dormant period. The $t\text{-BuP}_1/t\text{-BuP}_1\text{-H}^+$ complex corresponds to dormant species that can suppress the activation of $t\text{-BuP}_1\text{-H}^+$ to PO (Figure S47). After the complete consumption of TAz, the $t\text{-BuP}_1\text{-H}^+$ released from $t\text{-BuP}_1$ switches to activated species and starts the second copolymerization cycle (PA/PO ROCOP). Similar proton shuttling between the catalysts and growing chain end is considered key to promote selectivity and control for the copolymerization.⁴⁵ For both copolymerization cycles, the phosphazanium shows high monomer selectivity and polymerization controllability.

Figure S43. Plausible mechanistic pathway for $t\text{-BuP}_1$ -catalyzed terpolymerization of TAz, PA, and PO.

Figure 7. Plausible terpolymerization pathways of chemoselective TAz/PA ROCOP and PA/PO ROCOP by a simple organocatalyst, *t*-BuP₁.

REVIEWER COMMENTS

Reviewer #1 Made remarks to the editor only and is satisfied with the revisions

Reviewer #2 (Remarks to the Author):

The submission by Hadjichristidis and colleagues has been extensively reworked and much improved upon response to the original reviewer comments, and I am willing to recommend this manuscript for publication in Nature Communications. There are a few small changes that I would recommend prior to publication.

“Thermal analysis” may be more appropriate than “Thermodynamic analysis”?

Scheme S43 is useful but is difficult to read as it is very small – please can this be made larger.

The text added to address reviewer 3 point 9 is a little unclear - I would recommend rephrasing the sentence that was newly added to the manuscript. As the reactions were performed in triplicate, it would be helpful to add error values.

Reviewer #3 (Remarks to the Author):

After reviewing this work for the second time, it is obvious that the authors invested considerable experimental efforts in order to address issues that were brought up in the first reviewing process. Overall, the more technical matters were all addressed, including revisions of the catalytic cycles figures, addition of necessary references and emphasis of monomer ratios in the text. The authors performed an additional series of copolymerization reactions, varying the monomers ratio and employing different catalysts. The thermal analysis of the obtained copolymers was extended as well, now comprising of both DSC and TGA measurements (for which I would suggest the term “thermal properties/analysis” in the manuscript and SI instead of “thermodynamic analysis”) that provide a more whole picture of the affect of each segment on the glass transition and decomposition temperatures, as well as further evidence for the lack of phase separation of the two blocks. Nevertheless, the remaining concerns are still the novelty of the work and its universal nature. The authors only used different epoxides for the copolymerization, whereas the anhydride (PA) and N-sulfone aziridine (TAz) were not substituted with corresponding monomers. Since the authors reported that the perfect alternating selectivity in monomer enchainment is not retained upon the replacement of PO with a range of bulky epoxides, the universal nature of this process is still questionable. Variation of the aziridine and anhydride would have added important information on this aspect. In addition, there are still a couple of additional issues with this work:

1) Page 4: “the aromatics” should be “the aromatic signal” (or “peaks”), and four lines bellow: “...observed in the ¹H NMR spectra (Figure S1),.....”.

- 2) Page 4: "maintaining" should be "maintained" or "retained".
- 3) Page 6: The authors state that the "increased monomer concentration accelerates the ROCOP of TAz/PA. However, this is inaccurate as the concentration of TAz and PA is still the same (1 M) so the rate change is related to the medium and not the concentration, unless PO plays a non-innocent role in the first cycle as well. The authors should provide a better explanation for the rate change.
- 4) Page 13: The authors should explain in more specific terms how the NMR experiments support their mechanistic proposal. Such discussion could be added to the SI as well (in section 11), but the reader should not have to decipher the NMR titration results by himself instead.
- 5) Page 17: As pointed in my previous comments, the authors should better explain why the steric hindrance of the bulkier epoxides is responsible for the incomplete terpolymerization and a decrease in the alternating nature of Taz/PA (i.e., inefficient initiation and etc.)
- 6) Page 18: The text under the discussion section is actually the conclusion for this work, the discussion itself is in the "Results" section. Furthermore, in these conclusions the authors refer to the general ROCOP of N-sulfonyl aziridines/cyclic anhydrides, however as mentioned above this work only includes a single example of aziridine and anhydride, turning this general statement questionable.
- 7) Page 19: Unfortunately, the methods section still does not explain in enough details how the ¹H NMR monitoring of the copolymerization reactions (and acquisition of kinetic data) was performed experimentally. For example, how did the authors withdraw aliquots from a sealed reactor at 100 C without opening it to air? Were the reactions cooled down to R.T. before opening the reactor? In addition, the authors should emphasize in the experimental section that the process was performed in triplicates and provide the corresponding average and error values.
- 8) Page 20: The authors should be explicit about the use of a FTIR instrument equipped with a dip probe and describe the set up they used to monitor the reaction. Such description should be added to the representative general procedure.

Reviewer #4 (Remarks to the Author):

We have been asked to review the computational aspects of the current manuscript. While the computations seem to have been done with some confidence, the discussions and the descriptions need considerable work. As it is, the computations add almost nothing to the substance of the work save to make it longer, and we're not convinced that helps the work.

There are several aspects that are of some interest here that are completely not discussed.

For an example, on Figure 8 left, the TAz process has a higher barrier than the PA process, but the thermodynamics of the product are switched. That is interesting chemically, and will have meaningful effect on the polymers that form or may impact the polymerization process.

For another example, on Figure 8 right, the TAz process is lower in energy than the corresponding PO process. But N is less electronegative than O, so the opposite trend should be expected. Why is this different? Is it purely because the Ts group modifies the electronegativity of the N? Wouldn't model systems that probe this aspect add considerable value to the understanding of how electronics can impact reaction energetics?

Is the organocatalyst still attached to the basic nucleophile following protonation? Or is it completely ionic in nature? Which of the myriads of nitrogens perform the deprotonation? What does the coordination complex look like?

We are of the opinion that the experimental work does seem interesting and of general importance. However, the computational aspects are short-sighted and shallow. In particular, the role of the catalyst should be probed computationally - there is significant scientific information that the readership of Nature would be accustomed to expecting that is simply not there in this version of the manuscript.

Comments from the reviewers and our responses:

Replies to Reviewer(s)' Comments

Reviewers' comments:

Reviewer #1 (Remarks to the Author):

Reviewer #1 Made remarks to the editor only and is satisfied with the revisions.

Answer: Thank you very much for your support.

Reviewer #2 (Remarks to the Author):

1. The submission by Hadjichristidis and colleagues has been extensively reworked and much improved upon response to the original reviewer comments, and I am willing to recommend this manuscript for publication in Nature Communications. There are a few small changes that I would recommend prior to publication.

Answer: Thank you very much for your positive recommendation.

2. "Thermal analysis" may be more appropriate than "Thermodynamic analysis"?

Answer: Corrected to "Thermal analysis".

3. Scheme S43 is useful but is difficult to read as it is very small – please can this be made larger.

Answer: Supplementary Fig. 50 (original Figure S43) was improved.

4. The text added to address reviewer 3 point 9 is a little unclear - I would recommend rephrasing the sentence that was newly added to the manuscript. As the reactions were performed in triplicate, it would be helpful to add error values.

Answer:

We have already rephrased the newly added sentences in the method section as follows:

As an example, the procedure corresponding to entry 7 (Table 1) is given below. Eleven 100 mL dry Schlenk thick reactors were prepared and charged, in an argon-filled glovebox, with the same feed ratio, which is PA (0.264 g, 1.8 mmol), TAz (0.116 g, 0.6 mmol), initiator (N-benzyl-4-methylbenzenesulfonamide) (0.0052 g, 0.02 mmol), and anhydrous epoxides (0.6 mL) (ideal gas law was used to estimate internal pressure). After stirring for 5 min, t-BuP₁ (4.6 μL, 0.02 mmol) was added to each mixture. The reactors were sealed and then removed from the glovebox and stirred at 100°C to initiate the polymerization. At a specific time, a reactor was taken out of the oil bath and quickly cooled down in an ice-water bath. A sample was taken from the crude product for SEC and NMR analysis. The produced terpolymer was precipitated by the slow addition of the diluted reaction solution (CHCl₃) to an excess of methanol. The precipitation process was repeated three times, and the terpolymers were dried in a vacuum oven at 40°C overnight. Isolated yield = 0.40 g, 87% (before taking samples). The procedure was repeated three times and the average value, as well the standard error bars, are given in the graphs.

Another example of the procedure corresponding to entry 11 (Table 1) is the following. Eleven 100 mL dry Schlenk thick reactors were prepared and charged, in an argon-filled glovebox, with the same feed ratio, which is PA (0.264 g, 1.8 mmol), TAz (0.116 g, 0.6 mmol), and initiator (N-benzyl-4-methylbenzenesulfonamide) (0.0052 g, 0.02 mmol) (ideal gas law is used to estimate internal pressure). The reactors were placed in a freezer inside a glove box (-20°C) for 60 min. The reactors were removed from the freezer, and anhydrous EO (0.6 mL) was added to each reactor. After stirring for 5 min, the reactors were placed back in the freezer (-20°C) for 60 min, then *t*-BuP₁ (4.6 μL, 0.02 mmol) was added to each cold mixture. The reactors were sealed and then taken out of the glovebox and stirred at 100°C to initiate the polymerization. At a specific time, a reactor was taken out of the oil bath and quickly cooled down in an ice-NaCl bath. A sample was taken from the crude product for SEC and NMR analysis. The produced terpolymer was precipitated by the slow addition of the diluted reaction solution (CHCl₃) to an excess of methanol. The precipitation process was repeated three times, and the terpolymers were dried in a vacuum oven at 40°C overnight. Isolated yield = 0.39 g, 89% (before taking samples). The procedure was repeated three times, and the average value, as well the standard error bars, are given in the graphs.

The standard error bars were added in Figure 6.

Figure 6. (a) Plots of monomer conversions versus time at 100 °C, monitored by ¹H NMR spectroscopy. (b) Molecular weights ($M_{n,NMR}$) calculated by ¹H NMR and molecular weight distributions (\mathcal{D}) over time for the terpolymerization of TAz, PA, and PO. (c) Plots of monomer conversions versus time at 60°C, monitored by ¹H NMR spectroscopy and *in-situ* FTIR (error bars represent standard error).

Reviewer #3 (Remarks to the Author):

After reviewing this work for the second time, it is obvious that the authors invested considerable experimental efforts in order to address issues that were brought up in the first reviewing process. Overall, the more technical matters were all addressed, including revisions of the catalytic cycles figures, the addition of necessary references and emphasis of monomer ratios in the text. The authors performed an additional series of copolymerization reactions, varying the monomers ratio and employing different catalysts. The thermal analysis of the obtained copolymers was extended as well, now comprising of both DSC and TGA measurements (for which I would suggest the term "thermal properties/analysis" in the manuscript and SI instead of "thermodynamic analysis") that provide a more whole picture of the affect of each segment on the glass transition and decomposition temperatures, as well as further evidence for the lack of phase separation of the two blocks.

Answer: We thank the Reviewer for his careful review and suggestions.

We corrected the "thermodynamic analysis" to "thermal analysis" in the MS and SI.

Nevertheless, the remaining concerns are still the novelty of the work and its universal nature. The authors only used different epoxides for the copolymerization, whereas the anhydride (PA) and N-sulfone aziridine (TAz) were not substituted with corresponding monomers. Since the authors reported that the perfect alternating selectivity in monomer enchainment is not retained upon the replacement of PO with a range of bulky epoxides, the universal nature of this process is still questionable. Variation of the aziridine and anhydride would have added important information on this aspect.

Answer:

Despite the fact that we believe that our work is extremely demanding and innovative, we followed the constructive suggestion of the Reviewer and performed new experiments to prove the universal nature of our work.

This is the first time to synthesize pure diblock dialternating copolymers without any tapered sequences between the blocks (superior mechanical properties, ref. 44) from a mixture of three monomers without additional steps ("one-pot/one-step"). Besides, it is the first time that the complete absence of tapered sequences between blocks was proven using so many characterization techniques, including NMR monitoring, SEC traces, MALDI-ToF, H/C NMR spectra, DOSY spectra, (*in-situ*) FTIR, kinetic study, reactivity ratios, NMR titration, and DFT calculations.

Two new N-sulfone aziridines (BAz: N-brosylaziridine and NAz: N-(4-nitrobenzenesulfonyl) aziridine) were synthesized and used in terpolymerization with three different epoxides and one anhydride. Unfortunately, due to their sensitivity to the organocatalyst (*t*-BuP₁), the choice

of anhydrides is limited. The new experiments with the corresponding Figures were added in the MS and SI (Supplementary Table 14 and Table 15, Supplementary Figs. 30-36).

Two more *N*-sulfonyl aziridines were synthesized and used in the terpolymerization with PA and PO. The highly active *N*-brosylaziridine (BAz) and *N*-(4-nitrobenzenesulfonyl) aziridine (NAz) accelerate the first copolymerization stage (Supplementary Table 14 and 15).

7. Terpolymerization of PA, and PO with two more *N*-sulfonyl aziridines

Supplementary Table 14. Terpolymerizations of BAz, PA, and PO using *t*-BuP₁ as catalyst^a

Entry	Time	Conv. (BAz) ^b /%	Conv. (PA) ^b /%	$M_{n,theo}^c$ /kg mol ⁻¹	$M_{n,NMR}^b$ /kg mol ⁻¹	\bar{D}^d
1	2 min	77	26	9.74	9.28	1.06
2	5 min	99	33	12.5	11.7	1.05
3	10 min	99	33	12.5	11.7	1.05
4	15 min	99	33	12.5	11.7	1.06
5	30 min	99	33	12.5	11.7	1.08
6	1 h	99	46	14.9	14.8	1.09
7	2 h	99	56	16.8	16.9	1.10
8	3 h	99	75	20.3	20.7	1.12
9	4 h	99	91	23.3	23.6	1.11
10	5 h	99	99	24.7	25.1	1.10

^aThe terpolymerizations were performed at a ratio of [BAz]₀/[PA]₀/[BnN(H)Ts]₀/[*t*-BuP₁]₀ = 30/90/1/1 ([BAz]₀ = 1.0 M in PO) at 100°C. ^bDetermined by ¹H NMR in CDCl₃ using integrals of the characteristic signals. ^cCalculated as follows: (M.W. of Initiator) + ([BAz]₀/[I]₀) × conv.(BAz) × (M.W. of BAz + M.W. of PA) + {([PA]₀/[I]₀) × conv.(PA) - ([BAz]₀/[I]₀) × conv.(BAz)} × (M.W. of epoxides + M.W. of PA). ^dDetermined by SEC at 35°C in THF (1.0 mL·min⁻¹) using PSt standards.

Supplementary Figure 30. Stacked ^1H NMR spectra (400 MHz, CDCl_3 , 25°C) and SEC traces (THF, 35°C) of the reaction mixture at a ratio of $[\text{BAZ}]_0/[\text{PA}]_0/[\text{BnN}(\text{H})\text{Ts}]_0/[\text{t-BuP}_1]_0 = 30/90/1/1$ ($[\text{BAZ}]_0 = 1.0 \text{ M}$ in PO) at 100°C .

Supplementary Figure 31. ^1H NMR spectrum (400 MHz, CDCl_3 , 25°C) of poly(BAZ-*alt*-PA)-*b*-poly(PA-*alt*-PO).

Supplementary Figure 32. ^{13}C NMR spectrum (100 MHz, CDCl_3 , 25°C) of poly(BAz-*alt*-PA)-*b*-poly(PA-*alt*-PO).

Supplementary Figure 33. DOSY spectrum (600 MHz, CDCl_3 , 25°C) of poly(BAz-*alt*-PA)-*b*-poly(PA-*alt*-PO).

Supplementary Table 15. Terpolymerizations of NAz, PA and PO using *t*-BuP₁ as catalyst^a

Entry	Time	Conv. (NAz) ^b /%	Conv. (PA) ^b /%	$M_{n,theo}^c$ / kg mol ⁻¹	$M_{n,SEC}^d$ / kg mol ⁻¹	\bar{D}^d
1	2 min	99		11.6	7.79	1.08
2	5 min	99			7.79	1.06
3	10 min	99			7.79	1.06
4	15 min	99			7.79	1.05
5	30 min	99			7.79	1.06
6	1 h	99			9.36	1.09
7	2 h	99			16.8	1.09
8	3 h	99			20.9	1.14
9	4 h	99	99	23.7	20.2	1.12
10	5 h	99	99	23.7	20.2	1.11

^aThe terpolymerizations were performed at a ratio of $[NAz]_0/[PA]_0/[BnN(H)Ts]_0/[t-BuP_1]_0 = 30/90/1/1$ ($[NAz]_0 = 1.0$ M in PO) at 100°C. ^bDetermined by ¹H NMR in CDCl₃ using integrals of the characteristic signals. ^cCalculated as follows: $(M.W. \text{ of Initiator}) + ([NAz]_0/[I]_0) \times \text{conv.}(NAz) \times (M.W. \text{ of NAz} + M.W. \text{ of PA}) + (([PA]_0/[I]_0) \times \text{conv.}(PA) - ([NAz]_0/[I]_0) \times \text{conv.}(NAz)) \times (M.W. \text{ of epoxides} + M.W. \text{ of PA})$. ^dDetermined by SEC at 35°C in THF (1.0 mL min⁻¹) using PST standards.

Supplementary Figure 34. Stacked ¹H NMR spectra (400 MHz, CDCl₃, 25°C) and SEC traces (THF, 35°C) of the reaction mixture at a ratio of $[NAz]_0/[PA]_0/[BnN(H)Ts]_0/[t-BuP_1]_0 = 30/90/1/1$ ($[NAz]_0 = 1.0$ M in PO) at 100°C. (The low solubility of poly(NAz-*alt*-PA) leads to low NMR signal)

Supplementary Figure 35. ^1H NMR spectrum (400 MHz, CDCl_3 , 25°C) of poly(NAz-*alt*-PA)-*b*-poly(PA-*alt*-PO).

Supplementary Figure 36. ^{13}C NMR spectrum (100 MHz, CDCl_3 , 25°C) of poly(NAz-*alt*-PA)-*b*-poly(PA-*alt*-PO). (The low solubility of poly(NAz-*alt*-PA) leads to low NMR signal)

In addition, there are still a couple of additional issues with this work:

1) Page 4: "the aromatics" should be "the aromatic signal" (or "peaks"), and four lines below: "...observed in the ^1H NMR spectra (Figure S1),.....".

Answer: Corrected to "the aromatic peaks" and "...observed in the ^1H NMR spectra (Supplementary Fig. 1),.....".

2) Page 4: "maintaining" should be "maintained" or "retained".

Answer: Corrected to "maintained".

3) Page 6: The authors state that the "increased monomer concentration accelerates the ROCOP of TAz/PA. However, this is inaccurate as the concentration of TAz and PA is still the same (1 M) so the rate change is related to the medium and not the concentration, unless PO plays a non-innocent role in the first cycle as well. The authors should provide a better explanation for the rate change.

Answer: The concentration of TAz increased from 0.5 M ($[\text{TAz}]_0 = 0.5$ M in epoxides and THF) to 1.0 M ($[\text{TAz}]_0 = 1.0$ in epoxides) due to the removal of THF. To express more clearly, we revised the footnote of Table 1 and the corresponding descriptions in the MS and SI.

4) Page 13: The authors should explain in more specific terms how the NMR experiments support their mechanistic proposal. Such discussion could be added to the SI as well (in section 11), but the reader should not have to decipher the NMR titration results by himself instead.

Answer: Corresponding discussions were added to each stacked NMR spectra in the SI (in section 12.2).

We added the following descriptions in the caption of Supplementary Fig. 51 in the SI:

As shown in Supplementary Fig. 51, the methyl signals in $t\text{-BuP}_1$ shifted to the high field (green dots●), and the aromatic peaks in benzoic acid (BA) shifted to the low field (blue dots●) in the mixture of BA and $t\text{-BuP}_1$. In addition, the active hydrogen signal shifted from 12.51 to 10.76 ppm (red dots●). These shifts indicated that the $t\text{-BuP}_1$ can deprotonate the proton of BA. We concluded that the equilibrium tends to generate $\text{COO}^-/t\text{-BuP}_1\text{-H}^+$ due to the stronger basicity of $t\text{-BuP}_1$ than BA.

We added the following descriptions in the caption of Supplementary Fig. 52 in the SI:

As shown in Supplementary Fig. 52, we mixed equivalent *t*-BuP₁ and aziridine in toluene-*d*₈ in Young's NMR tube under argon. There was almost no shift for the *t*-BuP₁ in the absence/presence of TAz. The results indicated that the *t*-BuP₁ cannot abstract the proton from aziridine.

We added the following descriptions in the caption of Supplementary Fig. 53 in the SI:

Supplementary Fig. 46 a-c (same as Supplementary Fig. 52) demonstrated the proton transfer from BA to *t*-BuP₁. After the addition of TAz into the mixture of BA and *t*-BuP₁ in toluene-*d*₈ in Young's NMR tube under argon (Supplementary Fig. 53 d), the active hydrogen signals continuously shifted to the high field (red dots●, 12.51 to 10.76 to 9.78 ppm), indicating the progress of deprotonation from BA to *t*-BuP₁ to ring-opened TAz. After ring-opening of TAz, the proton in the *t*-BuP₁-H⁺ is transferred to the new chain-end, forming -N(Ts)H/*t*-BuP₁. The methyl signals in *t*-BuP₁ returned to the low field slightly (blue dots●, 2.38 to 2.41 ppm), but the chemical shifts was still smaller than the original *t*-BuP₁ (2.52 ppm). The chemical shifts indicated the fast proton exchange between the *t*-BuP₁ and *t*-BuP₁-H⁺.

We added the following descriptions below the Supplementary Fig. 54 in the SI:

Supplementary Fig. 54 a-c (same as Supplementary Fig. 52) demonstrated the proton transfer from BA to *t*-BuP₁. After mixing equivalent *t*-BuP₁ and PO in toluene-*d*₈ in Young's NMR tube under argon (Supplementary Fig. 54 d), there was almost no shift observed for the *t*-BuP₁ in the absence/presence of PO. The results indicated that the *t*-BuP₁ cannot abstract the proton from PO. After the addition of PO into the mixture of BA and *t*-BuP₁ in toluene-*d*₈ in Young's NMR tube under argon (Supplementary Fig. 54 e), the methyl and active hydrogen signals in *t*-BuP₁ shifted to the low field (2.380 and 1.326 to 2.382 and 1.332 ppm as well as 10.76 to 10.81 ppm), indicating hydrogen bonding between the *t*-BuP₁-H⁺ and PO.

5) Page 17: As pointed in my previous comments, the authors should better explain why the steric hindrance of the bulkier epoxides is responsible for the incomplete terpolymerization and a decrease in the alternating nature of TAz/PA (i.e., inefficient initiation and etc.)

Answer: The following sentences were added to the MS, which is the answer to incomplete terpolymerization of the bulkier epoxides:

The polarity of the epoxides, including dielectric constant and dipole moment, maybe the reason for the slow or incomplete dissociation of the chain-end in the second copolymerization cycles, resulting in inefficient initiation and the formation of incomplete block copolymers. The substitution groups on the epoxides may hinder the nucleophilic attack of the chain-end and lead to incomplete initiation.

6) Page 18: The text under the discussion section is actually the conclusion for this work, the discussion itself is in the "Results" section. Furthermore, in these conclusions the authors refer to the general ROCOP of N-sulfonyl aziridines/cyclic anhydrides, however as mentioned above this work only includes a single example of aziridine and anhydride, turning this general statement questionable.

Answer: According to the Nature Communications formatting instructions, we merge the conclusions part into the Results section.

We have added the additional experiments with more *N*-sulfonyl aziridines. For details, please see Reviewer 3, answer 2.

The general statement of cyclic anhydrides was corrected to phthalic anhydride in the conclusions.

7) Page 19: Unfortunately, the methods section still does not explain in enough details how the ¹H NMR monitoring of the copolymerization reactions (and acquisition of kinetic data) was performed experimentally. For example, how did the authors withdraw aliquots from a sealed reactor at 100 C without opening it to air? Were the reactions cooled down to R.T. before opening the reactor? In addition, the authors should emphasize in the experimental section that the process was performed in triplicates and provide the corresponding average and error values.

Answer: Please see Reviewer 2, answer 4.

8) Page 20: The authors should be explicit about the use of a FTIR instrument equipped with a dip probe and describe the set up they used to monitor the reaction. Such description should be added to the representative general procedure.

Answer:

We added the corresponding description in the method section in the MS.

In-situ Fourier-transform infrared spectroscopy (FTIR) study was conducted using a ReactIR 45m (Mettler Toledo) and equipped with a dip probe. The background was collected after an empty Schlenk flask was flame-dried under a vacuum. The probe was transferred into the argon-filled glovebox. The tip of the probe was immersed in the reaction mixture (the mixture prepared by the entry 7 procedure). The reactors were sealed and then taken out of the glovebox and stirred at 100°C to start the polymerization. The spectra were collected every 1 min, and each spectrum was scanned 256 times.

Reviewer #4 (Remarks to the Author):

We have been asked to review the computational aspects of the current manuscript. While the computations seem to have been done with some confidence, the discussions and the descriptions need considerable work. As it is, the computations add almost nothing to the substance of the work save to make it longer, and we're not convinced that helps the work.

Answer: We thank the Reviewer for his professional suggestions in the computational aspects. This work is an important work mainly on polymer synthesis and characterization. The DFT calculations are an auxiliary tool to compare the ring-opening reactivity of three monomers for explaining the polymerization sequence of different monomers in a one-pot/one-step procedure. The obtained free energy barriers of the three monomers attacked by the carboxyl anion/nitrogen anion are sufficient to indicate the ring-opening reactivity of the different monomers.

We modified the following sentence in the MS to clarify clearly:

The DFT calculations (Gaussian 16, BP86+D3 (BJ) DFT hybrid functional, def2-TZVPP basis set) was carried out to get further insights of different ring-opening reactivity of the three monomers...

There are several aspects that are of some interest here that are completely not discussed. For an example, on Figure 8 left, the TAz process has a higher barrier than the PA process, but the thermodynamics of the product are switched. That is interesting chemically, and will have meaningful effect on the polymers that form or may impact the polymerization process.

Answer: The phenomenon questioned by the Reviewer is not a very special case. It can also be found in the following works (*Macromolecules* 2021, 54, 2, 763; *J. Am. Chem. Soc.* 2016, 138, 4120). In our case, the lower free energy barrier of the PA process than the TAz process demonstrates that the PA was easier/faster to be opened than the TAz. The negative Gibbs free energy (ΔG) of the two products shows that two reactions can proceed spontaneously. The lower thermodynamic of the product of TAz indicates that the product of TAz is more stable. Because the delocalization of negative charge on the entire -NTs through resonance reduces the entire electronic density. A detailed explanation of the Ts group is shown in the next answer.

For another example, on Figure 8 right, the TAz process is lower in energy than the corresponding PO process. But N is less electronegative than O, so the opposite trend should be expected. Why is this different? Is it purely because the Ts group modifies the electronegativity of the N? Wouldn't model systems that probe this aspect add considerable value to the understanding of how electronics can impact reaction energetics?

Answer: The Ts group is a strong electron-withdrawing group. When the N atom has the Ts substituent, the electrons can delocalize on the entire –NTs group through resonance. The electron delocalization greatly reduces the original nucleophilicity and basicity of the N atom. Meanwhile, the electron delocalization stabilizes the negative charge of the entire –NTs group, causing the C–NTs bond to be easily cleaved. This is why the Ts group is widely used as a good leaving group in organic chemistry (*Ultrasonics Sonochemistry*, 2015, 26, 15; *Org. Biomol. Chem.*, 2021, 19, 713). It is well-known that the –O– and –N– are poor leaving groups. As reported in the literature, the nucleophilic reactivity of sulfonates (–OTs) are far more than the –O– unsubstituted by Ts group (more than $\sim 10^{10}$ - 10^{12}). (*Tetrahedron*. 63, 2007, 5103; *J. Am. Chem. Soc* 1985, 107, 5717). The nucleophilic reactivity of –NTs can be regarded as similar to that of –OTs. Therefore, the nucleophilic reactivity of –NTs should be more than the –O–.

Yes, the Ts group modifies the electronegativity of the N and stabilizes –NTs group. But, there are many factors that affect the nucleophilic substitution reactions, including the structure of the substrate, the ability to the leaving groups, the nucleophilic ability of nucleophiles, and the solvent. Hence, it is difficult to compare the nucleophilic reactivity of two different structures only by electron density.

Is the organocatalyst still attached to the basic nucleophile following protonation? Or is it completely ionic in nature? Which of the myriads of nitrogens perform the deprotonation? What does the coordination complex look like?

Answer: Phosphazene bases are classical superbases in organic synthesis (*Macromol. Rapid Commun*. 2018, 39, 1800485; *Angew. Chem. Int. Ed.* 1993, 32, 1361). The catalytic mechanism of the phosphazene bases is well-documented. The interaction between the protonated phosphazene and the basic nucleophile does exist. They are not completely ionic in nature. In our case, for sulfonamide/*t*-BuP₁, the dynamic equilibrium shifts more to sulfonamide/*t*-BuP₁, and for benzoic acid (BA)/*t*-BuP₁, the dynamic equilibrium shifts more to *t*-BuP₁-H⁺/BA⁻ (Supplementary Fig. 50). The structure of *t*-BuP₁-H⁺ was added in Supplementary Fig. 50. A higher electron density of N=P double bond than a single bond of P-N makes the N linking double bond perform the deprotonation. After the deprotonation, charge delocalization occurs in the *t*-BuP₁-H⁺ (*Angew. Chem. Int. Ed.* 1987, 26, 1167). The similar coordination complexes were shown in some papers (*Adv.Synth. Catal.* 2016, 358, 1110; *Angew. Chem. Int. Ed.* 1987, 26, 1167; *Angew. Chem. Int. Ed.* 1993, 32, 1361; *Chem. Ber.* 1994, 127, 2435). However, in this work, regardless of whether the sulfonamide or BA is protonated by *t*-BuP₁, the relative ring-opening reactivity sequence of the three monomers will not be changed by the nucleophile. Our calculations have already provided the related free energy barriers of the three monomers, which indicate the ring-opening reactivity sequence.

We are of the opinion that the experimental work does seem interesting and of general importance. However, the computational aspects are short-sighted and shallow. In particular, the role of the catalyst should be probed computationally - there is significant scientific information that the readership of Nature would be accustomed to expecting that is simply not there in this version of the manuscript.

Answer: We thank the Reviewer for his commendation of our experimental work. The computational aspects in the MS were used to quantify the ring-opening reactivity of the three types of cyclic monomers. The DFT calculations confirmed that TAz/PA ROCOP performs more easily than PA/EO ROCOP, and provides evidence for the synthesis of the diblock dialternating copolymers. The role of the catalyst is indeed very important in the terpolymerization. However, the addition of the catalyst (*t*-BuP₁ with 42 atoms) in the calculations only increases the workload and time and does not change the relative activity sequences of the three monomers. We believe that our calculations are sufficient to prove our proposed assumptions.

It is the first time that the complete absence of tapered sequences between blocks was proven using so many characterization techniques, including NMR monitoring, SEC traces, MALDI-ToF, H/C NMR spectra, DOSY spectra, (*in-situ*) FTIR, kinetic study, reactivity ratios, NMR titration, and DFT calculations. These efficient characterization methods are important, common, and expecting for the readership of Nature Communications. A comprehensive and in-depth study of DFT calculations is meaningful from the perspective of professional calculations, but the calculations in our work are sufficient to verify our claims.

REVIEWERS' COMMENTS

Reviewer #3 (Remarks to the Author):

The revised version (2nd revision) of this manuscript addresses all of the technical issues that were brought up in my previous revision, as well as consists of additional data showcasing the generality of the reported methodology (with the exception of variation of the anhydride). As such, this version is much more complete and offers the reader a better perspective of the reported achievements. Therefore, the only remaining issue is still the novelty of the reported work with respect to the relevant literature, as well as the authors' own previous work. The reported phenomenon wherein the kinetic copolymerization of TAz and PO is considerably faster than that of PA and PO is of interest. However, the attempt of the authors to present the non-tapered diblock sequence of a copolymer as a major novelty by comparing to previous work with a 10% tapered structure (Coates 2008) is problematic in the light of achievements in the field since then (Please refer to recent work by Prof. Charlotte Williams on switchable catalysis as an example). Therefore, I leave to the discretion of the editors the decision about whether the novelty of the work meets the demands of Nature Communications.

Reviewer #4 (Remarks to the Author):

We have read this manuscript and the author responses in detail, and we are of the firm opinion that the authors have failed to address any of the points. The computational section is lackluster by any standard, in our opinion, even compared to publications in far lower tier journals. It does not help the authors at all. We would strongly suggest removing Figure 8 completely and minimize the discussions. OR discuss the results in a way that befits the authorship of Nature Communications.

The authors point out that much of phenomena is already understood and does not need to be discussed in the manuscript. We agree with the authors - hence our original review that the computational part adds nothing to the manuscript and is cursory at best.

Much of the DFT result discussions in page 15 is so basic that it is difficult to find any contemporary publication in a respectable journal that has this lack of depth. This sections is written as if the computed results match experiments therefore it vindicates the experiment. There are NO insights discussed or revealed. Our original review comments were directed to the authors for them to improve this section, but clearly the authors feel that none of the detailed discussions are necessary because it is already published or known. Then why discuss these results at all? Either this is new and should be discussed to reveal the new, or it is not new and should be eliminated. There isn't a third option where results the authors believe are widely known or published elsewhere gets published here with NO new added insight or discovery.

The author's points about how the catalyst involvement in the computations is shallow and unwise. So the catalyst doesn't matter in the computational investigation of a catalytic reaction? Not only does the catalyst complex to the reagents, but also in the transition structure, and in the product structures. There are serious complications of steric and changes in charges in the remote regions in the transition structure that make this non-trivial. We have looked at these reactions, and the processes are not trivial and not simple.

It is further our opinion that manuscripts with significant flaws of this nature are unworthy of publication in high-impact journals. The rest of the work is of reasonable quality. So rather than trying to win this argument to get it published, we would advise to improve the manuscript.

—"After geometry optimizations and vibration analysis" - It should be "vibrational analyses".

Comments from the reviewers and our responses:

Replies to Reviewer(s)' Comments

Reviewers' comments:

Reviewer #3 (Remarks to the Author):

The revised version (2nd revision) of this manuscript addresses all of the technical issues that were brought up in my previous revision, as well as consists of additional data showcasing the generality of the reported methodology (with the exception of variation of the anhydride). As such, this version is much more complete and offers the reader a better perspective of the reported achievements. Therefore, the only remaining issue is still the novelty of the reported work with respect to the relevant literature, as well as the authors' own previous work. The reported phenomenon wherein the kinetic copolymerization of TAz and PO is considerably faster than that of PA and PO is of interest. However, the attempt of the authors to present the non-tapered diblock sequence of a copolymer as a major novelty by comparing to previous work with a 10% tapered structure (Coates 2008) is problematic in the light of achievements in the field since then (Please refer to recent work by Prof. Charlotte Williams on switchable catalysis as an example). Therefore, I leave to the discretion of the editors the decision about whether the novelty of the work meets the demands of Nature Communications.

Answer: Thank you for approving our revision. We fully understand your concern for the novelty of our work. Below we try to explain why our work is important.

In previous papers, a 10% tapered sequence between the pairs of blocks was inevitable, as the catalysts used lacked of selectivity, and the main driving force was the reactivity difference between the three monomers. Consequently, it was challenging to find a selective catalyst to avoid the presence of tapered sequences. To solve this problem, some works have used switchable catalysis, as summarized in a recent work by Prof. Charlotte Williams. However, switchable catalysis requires one-pot/two-step or two-pot/two-step methodologies.

In our case, we do not need to switch the catalyst and we believe that this is a significant development in the field. The catalyst automatically switches from the first-copolymerization to the second-copolymerization after one of the monomers involved in the first-copolymerization is completely consumed. The structure of the diblock dialternating copolymers without any tapered sequences was verified using all possible characterization methods.

Reviewer #4 (Remarks to the Author):

We have read this manuscript and the author responses in detail, and we are of the firm opinion that the authors have failed to address any of the points. The computational section is lackluster by any standard, in our opinion, even compared to publications in far lower tier journals. It does not help the authors at all. We would strongly suggest removing Figure 8 completely and minimize the discussions. OR discuss the results in a way that befits the authorship of Nature Communications.

The authors point out that much of phenomena is already understood and does not need to be discussed in the manuscript. We agree with the authors - hence our original review that the computational part adds nothing to the manuscript and is cursory at best.

Much of the DFT result discussions in page 15 is so basic that it is difficult to find any contemporary publication in a respectable journal that has this lack of depth. This sections is written as if the computed results match experiments therefore it vindicates the experiment. There are NO insights discussed or revealed. Our original review comments were directed to the authors for them to improve this section, but clearly the authors feel that none of the detailed discussions are necessary because it is already published or known. Then why discuss these results at all? Either this is new and should be discussed to reveal the new, or it is not new and should be eliminated. There isn't a third option where results the authors believe are widely known or published elsewhere gets published here with NO new added insight or discovery.

The author's points about how the catalyst involvement in the computations is shallow and unwise. So the catalyst doesn't matter in the computational investigation of a catalytic reaction? Not only does the catalyst complex to the reagents, but also in the transition structure, and in the product structures. There are serious complications of steric and changes in charges in the remote regions in the transition structure that make this non-trivial. We have looked at these reactions, and the processes are not trivial and not simple.

It is further our opinion that manuscripts with significant flaws of this nature are unworthy of publication in high-impact journals. The rest of the work is of reasonable quality. So rather than trying to win this argument to get it published, we would advise to improve the manuscript.

—"After geometry optimizations and vibration analysis" - It should be "vibrational analyses".

Answer: We agree with the reviewer and have removed the computational portion. A computational study in the presence of the catalyst, which will give an additional proof of our findings, takes much more time and it is beyond the scope of this work. In the future, we will try to carry out a detailed computational study.